# Exploring bias in OCO-3 Snapshot Area Mapping mode via geometry, surface, and aerosol effects

Emily Bell[1], Christopher W. O'Dell[1], Thomas E. Taylor[1], Aronne Merrelli[2], Robert R. Nelson[3], Matthäus Kiel[3], Annmarie Eldering[3], Robert Rosenberg[3], and Brendan Fisher[3]

[1]Cooperative Institute for Research in the Atmosphere, Colorado State University, Fort Collins, CO, 80521, USA.
[2]Department of Climate and Space Sciences and Engineering, University of Michigan, Ann Arbor, MI, 48109, USA.
[3]Jet Propulsion Laboratory, California Institute of Technology, Pasadena, CA, 91109, USA.

**Correspondence:** Bell (embell@colostate.edu)

**Abstract.** The Atmospheric Carbon Observations from Space (ACOS) retrieval algorithm has been delivering operational column-averaged carbon dioxide dry-air mole fraction ($X_{CO_2}$) data for the Orbiting Carbon Observatory (OCO) missions since 2014. The ACOS Level 2 Full Physics (L2FP) algorithm retrieves a number of parameters, including aerosol and surface properties, in addition to atmospheric $CO_2$. Past analysis has shown that while the ACOS retrieval meets mission precision requirements of 0.1-0.5% in $X_{CO_2}$, residual biases and some sources of error remain unaccounted for (Wunch et al., 2017; Worden et al., 2017; Torres et al., 2019). Forward model and other errors can lead to systematic biases in the retrieved $X_{CO_2}$, which are often correlated with these additional retrieved parameters. The characterization of such biases is particularly essential to urban- and local-scale emissions studies, where it is critical to accurately distinguish source signals relative to background concentrations (Nassar et al., 2017; Kiel et al., 2021). In this study we explore algorithm-induced biases through the use of simulated OCO-3 Snapshot Area Mapping (SAM) mode observations, which offer a unique window into these biases with their wide range of viewing geometries over a given scene. We focus on a small percentage of SAMs in the OCO-3 vEarly product which contain artificially strong across-swath $X_{CO_2}$ biases spanning several parts per million, related to observation geometry. We investigate the causes of swath bias by using the timing and geometry of real OCO-3 SAMs to retrieve $X_{CO_2}$ from custom simulated L1b radiance spectra. By building relatively simple scenes and testing a variety of parameters, we find that aerosol is the primary driver of swath bias, with a complex combination of viewing geometry and aerosol optical properties contributing to the strength and pattern of the bias. Finally, we seek to understand successful mitigation of swath bias in the new OCO-3 version 10 data product. Results of this study may be useful in uncovering other remaining sources of $X_{CO_2}$ bias, and may help minimize similar retrieval biases for both present missions (GOSAT, GOSAT-2, OCO-2, OCO-3, TanSat) and future missions (e.g. MicroCARB, GeoCarb, GOSAT-GW, CO2M).

## 1 Introduction

With the human-induced warming of Earth's climate system well underway, the study of anthropogenic greenhouse gas emissions - and in particular, carbon dioxide ($CO_2$) - plays a major role in the development of both local and international climate

policy. A robust understanding of Earth's carbon cycle, including both natural and anthropogenic contributions, is essential, and involves a myriad of challenges. Since the launch of the SCanning Imaging Absorption spectroMeter for Atmospheric CartographY (SCIAMACHY; Bovensmann et al., 1999) in 2002 aboard the European Space Agency's Envisat, space-based instruments have been addressing the particular challenge of scale: in decades prior, the global carbon cycle was studied using a handful of highly localized ground measurements scattered across, mostly, the northern hemisphere land surface; SCIA-MACHY and its successors have changed this limitation profoundly (Buchwitz et al., 2007; Schneising et al., 2008). The Greenhouse gases Observing Satellite (GOSAT; Kuze et al., 2009; Yokota et al., 2009) launched in 2009, as well as the Orbiting Carbon Observatory missions (OCO-2 and OCO-3), both launched in the 2010s, have improved upon SCIAMACHY's ability to measure $CO_2$ over a large fraction of Earth's surface, with more continuous spatial coverage than ground-based networks can provide. With their increasingly fine spatial resolution, precision, and accuracy, space-based observations from the OCO missions can now resolve carbon sources around the globe on scales as fine as those of individual power plant plumes (Nassar et al., 2017; Reuter et al., 2019).

OCO-3, a three-band grating spectrometer that measures reflected sunlight in the near infrared, was launched in 2019 and is mounted onboard the International Space Station (ISS). It observes Earth's atmosphere in eight across-swath footprints measuring roughly 1.6 by 2.2 kilometers squared on the ground. Just like its precursor OCO-2, it targets column-averaged carbon dioxide ($X_{CO_2}$) with spectral measurements of reflected sunlight in three bands: the oxygen-A band ($O_2$ A-band) at 0.76 micrometers, a weakly absorbing ("weak") $CO_2$ band at 1.61 micrometers and a more strongly absorbing ("strong") $CO_2$ band at 2.06 micrometers (Taylor et al., 2020). Best estimates of $X_{CO_2}$ are calculated via an optimal estimation method (Rodgers, 2000), using the Atmospheric Carbon Observations from Space (ACOS) Level 2 Full Physics (L2FP) retrieval algorithm, which has been shown to achieve OCO-2 mission precision requirements of 0.1-0.5% (Wunch et al., 2017; O'Dell et al., 2018). Operational data include a binary quality flag which uses several variables potentially indicative of compromised data fidelity - including aerosol optical depths (AODs), retrieved surface albedo, surface roughness, and differences from model estimates - to filter out "lesser quality" soundings (Osterman et al., 2020). An empirical bias correction is also included with operational datasets, which includes a footprint bias correction, a parametric bias correction, and a global scaling factor - all of which act to mitigate systematic biases within the ACOS retrieved $X_{CO_2}$ (O'Dell et al., 2018).

Of particular use in the analysis of fine-scale carbon emissions estimates are the OCO-2 and 3's Target and OCO-3's Snapshot Area Mapping (SAM) mode measurements. In Target and SAM modes, the instrument points at a specific off-nadir location and scans multiple times during an overpass, in an effort to produce a data-dense, spatially coherent map of $X_{CO_2}$. Details of these observation modes will be discussed in the next section. These measurements are the first of their kind to target urban- and local-scale emissions, such as those from megacities or individual power plants. Figure 1 provides an example of a visible $X_{CO_2}$ enhancement (or, rather, two areas of enhanced $X_{CO_2}$) over a power plant site, as seen by OCO-3. In this case, the enhancement extends to the northeast across four OCO-3 swaths, which we define as individual along-track scans. Each swath is eight footprints wide - these are visualized in the right-hand panel as white rectangles. Point source signals are difficult

to quantify because the instrument noise is a similar order of magnitude to the $X_{CO_2}$ enhancement; the $X_{CO_2}$ enhancement is also often two orders of magnitude smaller than the background concentration. Nitrogen dioxide ($NO_2$), co-emitted with $CO_2$ in fossil fuel combustion, is a helpful validation source for fossil signals due to $NO_2$'s short lifetime and high concentration relative to background values. Indeed, co-located $NO_2$ observations have been shown to be helpful in plume identification when using OCO-2 data Reuter et al. (2019). We show observations from the Tropospheric Monitoring Instrument (TROPOMI) $NO_2$ product (Veefkind et al., 2012; Van Geffen et al., 2019) in the right-hand panel of Figure 1. Because the OCO-3 $X_{CO_2}$ and TROPOMI $NO_2$ observations compare so well, we believe the $X_{CO_2}$ enhancement to be a real feature of the atmospheric state in this scene.

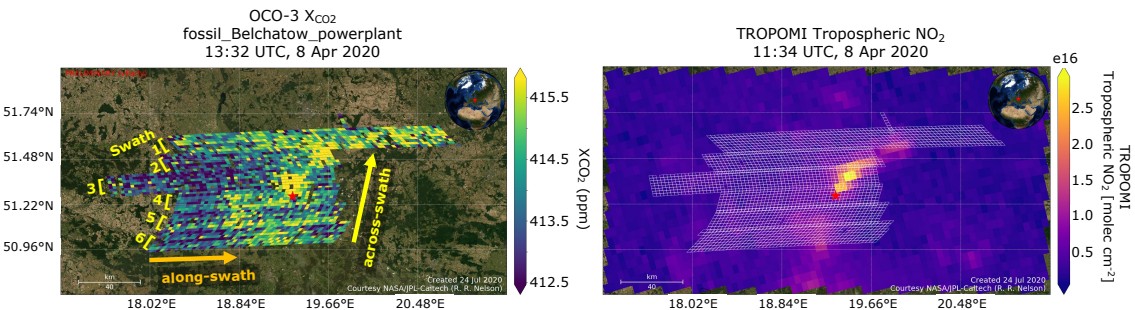

**Figure 1.** OCO-3 vEarly SAM showing observed $X_{CO_2}$ enhancements compared to coincident TROPOMI $NO_2$ data over Bełchatów power station in Poland. We have identified individual swaths using brackets, and the "along-swath" and "across-swath" dimensions, which we will refer to repeatedly in this work, are designated by the colored arrows.

A clear distinction between $X_{CO_2}$ enhancement and background is critical, and can be easily complicated by retrieval biases stemming from a variety of factors. Through small-area analysis of OCO-2 Target data, it is well-known that estimates of $X_{CO_2}$ from the ACOS L2FP retrieval algorithm have errors dependent on aerosol and clouds within the field of view, variations in the surface reflectances and Bi-directional Reflectance Distribution Functions (BRDF), viewing and solar geometries (Wunch et al., 2017; Worden et al., 2017; Torres et al., 2019). Target and SAM mode measurements are especially helpful for evaluating geometry-related effects, as they take a large number of measurements (typically several thousand soundings) over less than a two-minute time period: they sample a near-constant atmospheric state, leaving the changing geometry as the primary independent variable. Thus, in both Targets and SAMs, we expect any spatially coherent biases to be due primarily to the retrieval's imperfect treatment of the effects of changing solar and observation geometry, or changes in surface albedo within the scene.

SAM measurements are novel in their spatial coverage, specific to OCO-3, and are a valuable resource in studying geometry-related effects. In fact, the OCO-3 vEarly dataset - the first publicly released OCO-3 L2FP product - provides a striking operational example of geometry-dependent biases in a small fraction of SAM cases. In these SAMs, we observe a highly unphysical $X_{CO_2}$ gradient of several parts per million (ppm) in the across-swath direction, often with a stepwise increase from swath to

swath: Figure 2 provides two examples. We refer to this phenomenon as "swath bias." Swath bias was found in multiple SAMs each month throughout vEarly data processing, and appeared to occur more frequently over sites with high surface albedo and scenes with high AODs. The swath bias is also regularly seen over urban sites, and the $X_{CO_2}$ gradient in these cases often meets or exceeds the average 1-5 ppm enhancement we expect from fossil fuel signals. This can make the differentiation between fbias and real signal quite challenging, and renders these SAM data less usable for emissions studies; thus, the magnitude, prevalence, and importance of OCO-3 swath bias SAMs make them an intriguing source of study. Future missions with similar GHG-monitoring strategies, such as MicroCARB (Pasternak et al., 2017; Bertaux et al., 2020), GeoCarb (Moore III et al., 2018; Nivitanont et al., 2019), GOSAT-GW (Kasahara et al., 2020), and CO2M (Ciais et al., 2017; Janssens-Maenhout et al., 2020), may benefit from an improved understanding of these types of biases.

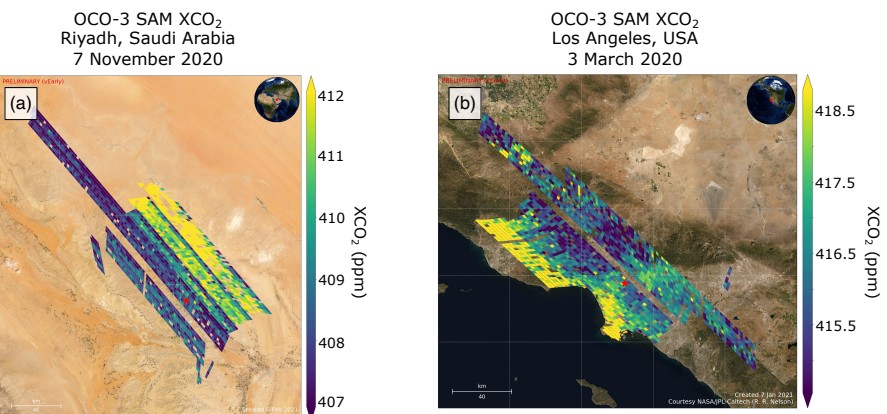

**Figure 2.** Two examples of "swath bias" from the raw (no filtering or bias correction) OCO-3 vEarly $X_{CO_2}$ data product, over (a) Riyadh, Saudi Arabia and (b) Los Angeles, USA.

In Sect. 2, we review the similarities and differences between OCO-3 Target and SAM mode measurements. Sect. 3 focuses on the swath bias (shorthand "SB") in the OCO-3 vEarly dataset, where we have created a set of criteria to identify a SB pattern in any given vEarly SAM. We identify SB in approximately 12% of vEarly SAMs, and identify potential systematic behavioral differences between those 12% of L2FP retrievals and the rest of the dataset. Optical path length, AOD, and surface albedo arise as particular variables of interest.

Simulation-based studies, such as O'Dell et al. (2012), are useful for identifying major sources of error in retrievals, such as those due to aerosol, clouds, or surface effects. They can also help to identify cause and effect rather than pure correlation. We therefore pursue a simulation-based approach to help identify the root causes of swath bias relative to the hypotheses we form in Sect. 2. We employ the L1b simulator (O'Brien et al., 2009) and the ACOS L2FP retrieval algorithm (O'Dell et al., 2012; O'Dell et al., 2018) to test those behaviors. In Sect. 4, we describe our simulations of OCO-3 SAM mode data,

starting from time and geometry information collected on-orbit. Semi-realistic scenes are built using information from the National Centers for Environmental Prediction (NCEP; Saha et al., 2014) model (meteorology and trace gases), CarbonTracker's CT2019B ($CO_2$; Jacobson et al., 2020), Cloud-Aerosol Lidar with Orthogonal Polarisation (CALIOP, vertical aerosol number concentrations; Winker et al., 2007), and Moderate Resolution Imaging Spectroradiometer (MODIS, surface albedo and BRDF parameters; Schaaf and Wang, 2015). We keep surface and aerosol setups relatively simple to isolate the effects of changing viewing geometry. We provide a brief overview of the ACOS retrieval algorithm in Sect. 5. Sect. 6 discusses the results of our full simulation experiments, from the generation of L1b spectra to the final retrieved L2FP $X_{CO_2}$ concentration. We initially attempt to replicate the observed SB in a few notable SAMs, and then focus on a set of three cases over one site. We manipulate different scene inputs - aerosol optical depth/type/height, and surface albedo - to evaluate their effect on the resulting simulated $X_{CO_2}$ patterns.

Finally, in Sect. 7 we seek to explain why the SB is significantly mitigated in the OCO-3 version 10 data product compared to vEarly. Ultimately, we hope that an improved understanding of the SB, summarized in Sect. 8, can prevent similar biases in future $CO_2$ monitoring missions.

## 2   OCO Target and SAM mode measurements

Both OCO-2 (Crisp et al., 2017; Eldering et al., 2017) and OCO-3 (Basilio et al., 2019; Eldering et al., 2019; Taylor et al., 2020) are well documented missions in the literature; finer details on instrumentation and global datasets will be left to other publications. Of specific importance to this work are the aforementioned OCO-3 SAM mode measurements, a successor to OCO-2's Target mode. Target mode was developed for OCO-2 to scan a small area continuously in an overlapping pattern as the satellite passes overhead. On OCO-2, this continuous scanning is achieved by changing the orientation, and thus viewing angle, of the satellite itself, through a series of complex maneuvers in-orbit. The pointing method of OCO-3, however, is mechanistically different, due to its fixed-mount position onboard the International Space Station (ISS). Rather than adjust the physical orientation of the instrument itself, OCO-3 is outfitted with a 2-D pointing mirror assembly (PMA), which provides the ability to scan larger areas during a single overpass. This broader scan defines OCO-3's SAM mode. A typical SAM spans a ground area of about 80x 80 km (1600 $km^2$), and each of the approximately 5 scans, or swaths, are adjacent to one another, rather than overlapping, to create a larger spatial "map" of $X_{CO_2}$ than OCO-2 Target mode. OCO-3 has recorded more than 7000 SAMs between August 2019 and June 2022. The OCO-3 SAM site list includes volcanoes, cities, power plants, and occasionally other scientific points of interest, such as flight campaigns or ground-based towers. Figure 3 gives a side-by-side example of an OCO-2 Target versus an OCO-3 SAM over Los Angeles, taken two years apart; Figure 4 illustrates the diversity of OCO-3 SAM and Target sites observed from launch to July 2022.

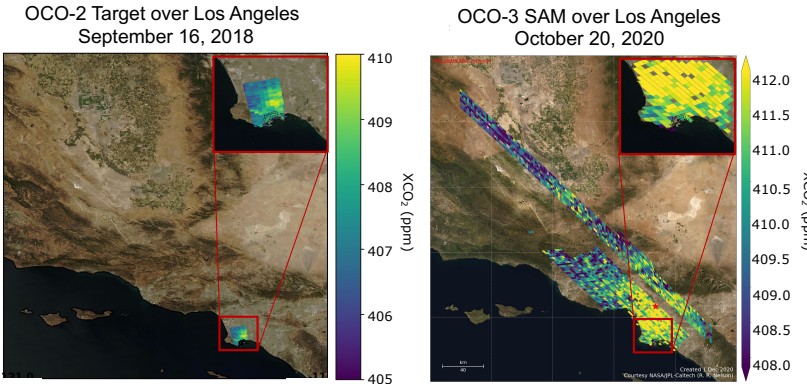

**Figure 3.** Visual comparison of Target and SAM mode observations over the LA basin. The left panel and insets were generated by overlaying OCO-3 data on RGB images from NASA Worldview.

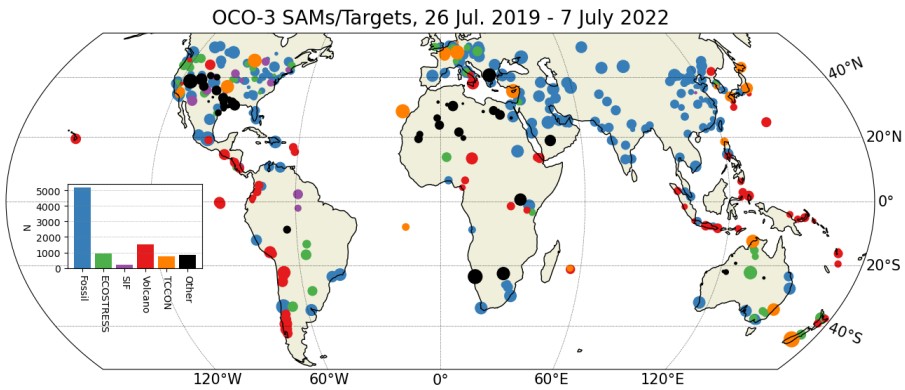

**Figure 4.** Map of all OCO-3 SAMs and Targets recorded through July 2022, categorized by site type.

OCO-2 Target mode is primarily used for validation purposes, with many targets located at Total Column Carbon Observing Network (TCCON; Wunch et al., 2011) sites, but it also includes targets over certain megacities and some point sources, such as volcanoes and large power plants. OCO-3's Target mode is utilized similarly, but has a larger spatial coverage which can also be used to study the carbon cycle on local scales (Kiel et al., 2021; Rißmann et al., 2022). Studies using OCO-2 Target data have shown the instrument's ability to detect anthropogenic $X_{CO_2}$ enhancements on subcontinental scales (Hakkarainen et al., 2016), from megacities (Schwandner et al., 2017), and from industrial sources such as iron and steel plants (Wang et al., 2018). Nassar et al. (2017) even used OCO-2 nadir and glint observations fitted to a Gaussian plume model to quantify $CO_2$ emissions from seven coal power plants, agreeing to within 1 to 17% of EPA estimates for sites in the US. In a follow-up study, Nassar et al. (2021) showed that averaging multiple overpasses of the same site can improve annual emissions estimates by both reducing random errors and addressing temporal variations. Reuter et al. (2019) similarly showed the value of combining OCO-2 observations of power plant plumes with $NO_2$ observations from the Sentinel-5 Precursor over six sites, comparing

their estimates of cross-sectional fluxes to existing emission inventories successfully within their uncertainties. Such successful implementation of nadir and glint mode data has motivated the use of Target and SAM data for similar studies: Kiel et al. (2021) utilized OCO-3 SAM and Target measurements to evaluate $X_{CO_2}$ concentrations over Los Angeles, and found good agreement with coincident TCCON $X_{CO_2}$ measurements, TROPOMI $NO_2$ estimates, and model emissions estimates. In any emissions study, separating the enhancement from the background is key - Nassar et al. (2017) found that the background $X_{CO_2}$ concentration was one of the main uncertainties in their work. Kiel et al. (2021) found that OCO-3 SAM mode measurements are able to sample both urban enhancements and background concentrations within the same overpass, so if we maintain a high fidelity of Target, and, with OCO-3, SAM data, a single Target or SAM can provide accurate information on both the enhancement and the background.

The presence of instrument and retrieval biases reduces the information content, however, especially on small spatial scales and for enhancements as small as those seen from individual point sources. OCO-2 and OCO-3 both suffer from aerosol-related biases: relying on reflected sunlight, they cannot fully distinguish between photons reflected by Earth's surface and those reflected by intermediate scatterers along the light path. The resultant changes in path length from intermediate scattering by aerosols can lead to erroneous $X_{CO_2}$ values (see e.g. Aben et al., 2007; Butz et al., 2009; O'Dell et al., 2012). In Target and SAM modes, especially, the misattribution of path length can be exacerbated by the quickly changing viewing geometry. In addition to aerosols, Wunch et al. (2017) and Worden et al. (2017) both evaluated OCO-2 data on regional scales to find that residual biases of up to 1.5 ppm remained compared to colocated TCCON sites, and that some residual noise in OCO-2 data may be due to variations in surface properties or solar zenith angle. It was this type of small area analysis which led to the earliest conception of a simulation-based investigation of geometry and aerosol-dependent biases, which we utilize in this study. The SB effect in vEarly, discussed in the previous section, may be related to any of these factors: aerosol, viewing and solar geometry, or even surface properties. While we focus our study on a robust evaluation of OCO-3 SAMs, we have also observed SB in some OCO-3 Targets after separating their overlapping scans. Given that the same principle of changing geometry over a constant atmospheric scene applies, we argue that the causes of SB in OCO-3 SAMs are also applicable to OCO-3 Targets.

## 3  Swath Bias in vEarly

In the early processing of OCO-3 SAM data, multiple SB cases were identified with each proceeding month. Initially, case identification was done by eye, inspecting plots of the data as it was processed. The magnitude and spatial coherence of the observed $X_{CO_2}$ biases made them the subject of curiosity and concern. The vEarly dataset includes nearly 6000 SAMs, spanning a wide variation of surface types, aerosol scenes, and solar and viewing geometries. To quantitatively identify what fraction of this collection suffers SB, we develop a SB "flag" using the following criteria:

1) the SAM must include at least 500 soundings,

2) the SAM must include at least four swaths with at least 100 soundings apiece (SAMs typically include 4 to 6 swaths), and

3) the ratio of the standard deviation of the in-swath $X_{CO_2}$ medians to the mean of the in-swath standard deviations is greater than 0.75. Mathematically:

$$\text{sb\_ratio} = \frac{\text{stddev(swath } X_{CO_2} \text{ medians)}}{\text{mean(swath } X_{CO_2} \text{ stdevs)}} = \frac{\text{sd\_median}}{\text{mean\_sd}} > 0.75 \tag{1}$$

In the rest of this study, we will refer to the quantity in the numerator as sd_median, and the denominator as mean_sd.

Essentially, the sb_ratio is a form of the "coefficient of variation," which quantifies the variability in across-swath $X_{CO_2}$ in relation to the mean of the population variability. The sb_ratio requires that for cases free of SB (sb_flag = 0), the noise across swaths must represent less than 75% of the noise within the scene. If the noise across swaths represents more than 75% the noise within the scene, we consider the scene the subject of a SB effect, and it is assigned a sb_flag of 1. This simple flag provides a computationally efficient means of evaluating all 5940 vEarly SAMs for the presence of SB. We acknowledge that this interpretation of sb_ratio assumes that any sufficient across-swath $X_{CO_2}$ variability is due specifically to SB, meaning that real $X_{CO_2}$ signals or other biases are negligible; this leaves room for error in our interpretation of the sb_flag, which we explore later in the Version 10 data product in Sect. 7.

The filtering process for the OCO missions in general involves two stages: pre-processing, which eliminates soundings prior to the L2FP retrieval (also called "sounding selection"), and post-processing, applied to the retrievals themselves. The first involves the A-Band preprocessor (ABP), which is used to retrieve surface pressure and is sensitive to clouds and aerosols (Taylor et al., 2016). Retrieved surface pressures whose difference from prior estimates (dP_abp) are anomalously large are assumed to be contaminated by clouds/aerosols, and are considered lower quality. A secondary pre-processing cloud filter is also applied, which relies on the ratio between strong and weak band retrieved $CO_2$ in the IMAP-DOAS preprocessor (IDP Taylor et al., 2016). Only sounding which pass both ABP and IDP are processed through L2FP. In post-processing, OCO-2 and OCO-3 Lite products include a quality flag to be applied by the user, which we describe briefly in Sect. 1, with further details of vEarly quality flag construction in Osterman et al. (2020).

For our vEarly analysis, we apply a simple post-processing quality filter to individual soundings prior to calculating sb_ratio. Review of the operational quality flag in vEarly shows that an increasingly unrealistic fraction of SAMs are flagged as "bad" (quality flag of 1) due to progressive calibration errors over the vEarly record. In the sounding selection process, a | dP_abp | < 30 filter is used to eliminate cloudy soundings before pushing clear" soundings through to L2FP (Taylor et al., 2016). We choose in this study to use an additional post-processing simple filter of | dP_abp | < 16 hPa to define "good" quality soundings. A similar range of dP_abp is typically used in the development of the operational quality flags (Taylor et al., 2020). This alone acts as a fairly relaxed filter, but it is critical for our investigation to retain enough soundings to see the swath bias. For the rest of this study, we will refer to this as the dP_abp quality filter.

The collection of 5940 vEarly SAMs, spanning August 2019 to June 2021, includes 2940 cases with 500 or more soundings (N>500) after the dP_abp filter is applied. Based on Equation 1, 352 of those cases are flagged as having SB - about 12%. This is a fairly significant fraction of the full dataset, not to mention that 256 of the 352 SB SAMs are urban/fossil sites, where
accuracy over small areas is particularly important for fine-scale emissions estimates; identification of bias and uncertainty is therefore essential. In fact, we find that SAMs over urban/fossil sites experience SB in nearly 14% of N>500 cases, whereas only 9% of non-fossil N>500 SAMs experience SB.

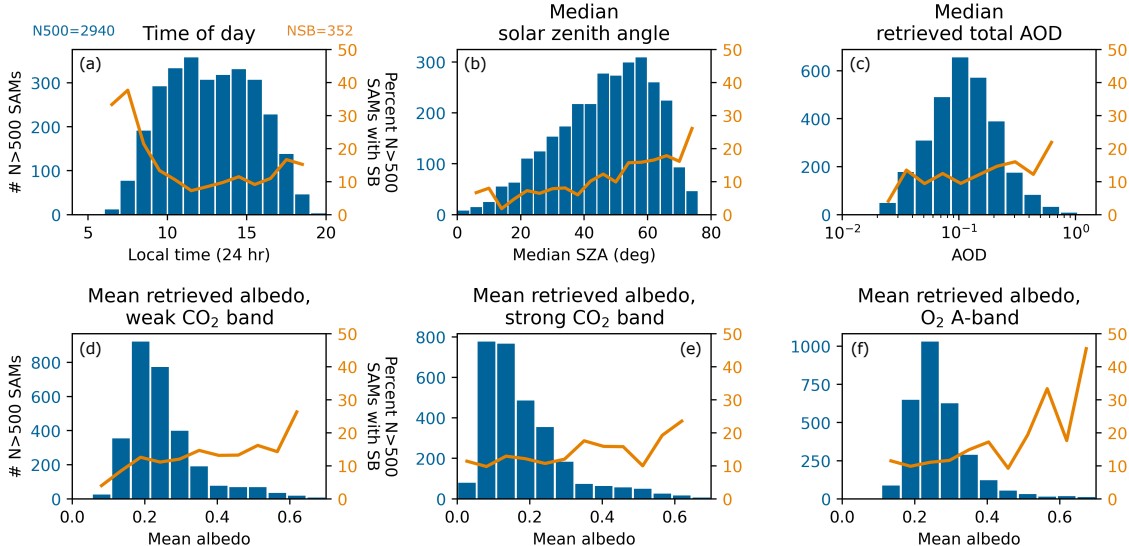

**Figure 5.** Histograms showing the distribution of various retrieval parameters across 2940 vEarly SAMs with 500 soundings or more. The orange line indicates the percentage of SAMs in each bin which have been flagged as having SB. This percentage was only calculated for bins containing at least ten SAMs.

There are a number of retrieved parameters which can help diagnose correlating factors of the SB within vEarly. We provide
histograms of several of these parameters in Figure 5. The N>500 SAM collection is shown in blue, and the percentage of N>500 SAMs with SB is shown in orange. A few trends are of interest here. A larger fraction of SAMs have SB in the morning hours, with a minimum around local noon. We also see a consistently larger percentage of SB SAMs at higher solar zenith angles. These two observations combined can be interpreted as a higher frequency of SB cases when scattering effects are stronger, e.g., longer slant paths through the observed atmospheric column. A longer path length means higher aerosol optical
depths, and in particular, we suspect aerosol optical depths to be of import, based on both Figure 5c and the historical difficulty of characterizing and accounting for aerosol effects within the ACOS algorithm (Worden et al., 2017; Wunch et al., 2017; Nelson and O'Dell, 2019). This hypothesis aligns with the fact that so many of our SB cases are at urban/fossil sites, which tend to be polluted.

Trends in the mean retrieved albedo also show higher fractions of SAMs at higher albedos, particularly in the $O_2$ A-band. This analysis confirms our early subjective observations, which pointed us toward scenes with brighter surfaces and higher AODs.

## 4   Simulation setup

Motivated by the results in the previous section, we turn to realistic simulations to systematically evaluate the sensitivity of algorithm-induced swath bias to parameters such as AOD and surface albedo. We begin with the actual instrument time and geometry information, determine realistic atmospheric profiles based on various model inputs, and finally calculate the of top-of-atmosphere radiance spectra. These simulations are variants of the simulation system described in O'Brien et al. (2009).

### 240   4.1   Generating atmospheres

    The first step in the simulation process is to create a simulated atmosphere. We use the real OCO-3 geometry data from the SAM of interest, to obtain location information such as date and time, latitude and longitude, and ISS/PMA/instrument and solar geometry.

NOAA's CarbonTracker version 2019B (CT2019B), is used for atmospheric carbon data (Jacobson et al., 2020). Native CT2019B $CO_2$ mole fractions are provided globally at 3-hourly, 3 degree longitude by 2 degree latitude resolution, with 35 vertical layers. Data is available through the end of 2018; in this study, we simulate three SAMs taken in 2020, using CT2019B data from the relevant day in 2018. Any mean offset in our simulated $X_{CO_2}$ values compared to the observed OCO-3 data can thus be attributed to the annual growth rate since 2018 - around 2.4 ppm/year (Tans and Keeling, accessed 15 June 2022).
We have not accounted for this offset in our simulations, as our investigation hinges on changes in small-scale spatial patterns rather than global mean increases.

    We use NOAA's National Centers for Environmental Prediction (NCEP) reanalysis for meteorological data. NCEP near-real-time data is available at 6-hourly global coverage on a 2.5 x 2.5 degree grid, and 17 pressure levels (Saha et al., 2014). We
apply a hypsometric surface pressure adjustment, and we use the specified month and day from 2018, as with CarbonTracker. The resampled CT2019B and NCEP reanalysis data are then combined with MODIS surface reflectances to build our final meteorology and scene datasets. The MCD43A1 MODIS BRDF/Albedo product used here is available globally at 500m resolution daily (Schaaf and Wang, 2015).

Our next step is to build a simple yet semi-realistic aerosol and surface scene. For a given SAM, we only include one aerosol type at a time, and always exclude water and ice clouds. We keep the specified vertical aerosol profile uniform across the

scene. Simulated aerosol types are prescribed in the L1b simulator framework. A full list of the aerosol types available in the simulator can be found in O'Brien et al. (2009), but in this study, we only utilize two: a coarse "dust" type, and a fine "clean continental" type. Aerosol effective radius, single scattering albedo, and other optical properties are derived from Dubovik et al. (2002). These aerosol types are different than those used in the ACOS L2FP retrieval (discussed in Sect. 5). To construct our profiles, we begin with an existing aerosol profile from a resampled Cloud-Aerosol Lidar with Orthogonal Polarisation (CALIOP) 05kmALay monthly field (Winker et al., 2007). The parameters for the selection of this initial profile are loose, because once it is chosen, we can change the aerosol type, adjust the layer up or down to any native CALIPSO vertical pressure level, and then scale the number densities to achieve our specified optical depth. Previous studies have shown radiances to be relatively insensitive to the geometric thickness of the aerosol layer (Butz et al., 2009; Frankenberg et al., 2012), so we do not control for this; it is determined only by the pressure layer thickness at the specified height. The mechanism used to adjust the vertical height of the aerosol layer puts the top of the layer at the native CALIPSO pressure level closest to specified pressure: for example, if a pressure of 800 hPa is input, the top of the aerosol layer may actually sit at 793 hPa. The exact value varies slightly based on the date and surface pressure, so for simplicity, we will refer to the approximate input heights, e.g., 750, 800, or 900 hPa. We apply our single profile, along with its associated surface elevation and surface reflectivity, to every sounding in the SAM.

## 4.2 Generating L1b radiances

Simulated radiances are generated by a forward model that has been used previously for simulation studies of GOSAT, OCO-2, and OCO-3, such as in Eldering et al. (2019). The radiative transfer module uses the specified aerosol, physical, and surface properties as inputs, along with geometry information. In this study, we calculate the gas absorption optical depths using the absorption coefficient lookup table ABSCO v5.1 (Payne et al., 2020). Options for surface treatment include a specified Lambertian albedo, a MODIS-derived Lambertian albedo, and a MODIS-derived BRDF albedo. We use the first when testing specific surface albedos in the three bands, and we use the last when testing aerosol properties with realistic albedos. We do not account for changes in surface topography, i.e., a fixed elevation is used for all soundings within a scene. Rayleigh scattering is calculated, as well as the solar spectrum, derived from a solar model described in Bösch et al. (2006). The solar spectrum is Doppler-shifted to account for the reference frame of the instrument; then, we obtain a "measured" spectrum via an instrument model, which convolves the instrument line shape (ILS) function with the simulated Doppler-shifted spectrum. We do not add instrument noise in this work for clarity, but doing so would add a component of random error to the radiances, and hence to the retrieved $X_{CO_2}$ (Connor et al., 2016).

## 5 ACOS L2FP retrieval algorithm

The ACOS algorithm is well-documented (O'Dell et al., 2012; O'Dell et al., 2018); here we provide a brief summary. For the simulated data, as with operational OCO data, predominant cloud screening is performed by the $O_2$-A band preprocessor (ABP), documented in Taylor et al. (2016). The A-band provides a means of accurate surface pressure retrieval, and uses the difference between retrieved and prior surface pressure (dP) to determine the presence of clouds.

After filtering by the ABP, the selected soundings are run through the L2FP retrieval. The retrieval utilizes an optimal estimation approach, iteratively minimizing a cost function to produce the most likely observed radiances, and ultimately, an estimate of $X_{CO_2}$. The L2FP state vector also includes a posterior $CO_2$ profile, aerosol optical depths, surface albedos, and surface pressure, among several other variables. Spectra generated by the retrieval utilize a radiative transfer model similar to the L1b simulator, which is described in the previous section. A full description of the ACOS retrieval algorithm is available in O'Dell et al. (2012). The OCO-3 vEarly dataset was produced using version 10 of the ACOS algorithm.

The L2FP algorithm used in our simulation study is v8 (O'Dell et al., 2018), but with the use of the newer ABSCO v5.1.0 (Payne et al., 2020) in order to match the simulator spectroscopy. Recall that ACOS v9 did not include any changes to the L2FP retrieval itself, and was a reprocessing of the L2 Lite files only, mainly to account for corrections in the OCO-2 pointing (Kiel et al., 2019). At the start of this work, the ACOS v10 L2FP was still in development. The retrieval state vector includes five aerosol types (O'Dell et al., 2018): always retrieved are water cloud, ice cloud, and a stratospheric aerosol; the remaining two aerosols are fixed types from the Modern-Era Retrospective analysis for Research and Applications 2 (MERRA2, Gelaro et al., 2017), and vary based on the time and location of the SAM. Prior meteorology comes from GEOS5 FP-IT (Rienecker et al., 2008), and is used for temperature profiles, water vapor profiles, and surface pressure.

## 6 Simulation experiments

Here we discuss the case selection and results of our simulated SB experiments. In this introduction, we test two SB SAMs from vEarly to see whether we can recreate the observational SB using simulated data. We then focus on three SAMs over a single representative target location, and use their geometry as templates to test the SB in a more complete scene state space: we manipulate each SAM to include various aerosol types, heights, and optical depths with a constant surface elevation and reflectivity. Finally, we hold aerosols constant while testing changes to surface albedo in all three bands.

To begin our investigation, we choose two desert scenes over Australia and Iran where the vEarly operational retrieval indicates a SB. Maps of several vEarly retrieved parameters for both SAMs are shown in Figures 6 and 7, with no filtering, in order to maximize comparison to simulations. In both figures, SB patterns in $X_{CO_2}$ - panel (b) - are primarily correlated to changes in the viewing geometry, which we represent in the two left panels. Panel (a) shows the sensor zenith angle, and (b)

325   shows the PMA elevation angle, a measurement of the geometry of the pointing mechanism, where positive angles indicate forward-looking observations and negative angles indicate backward-looking observations.

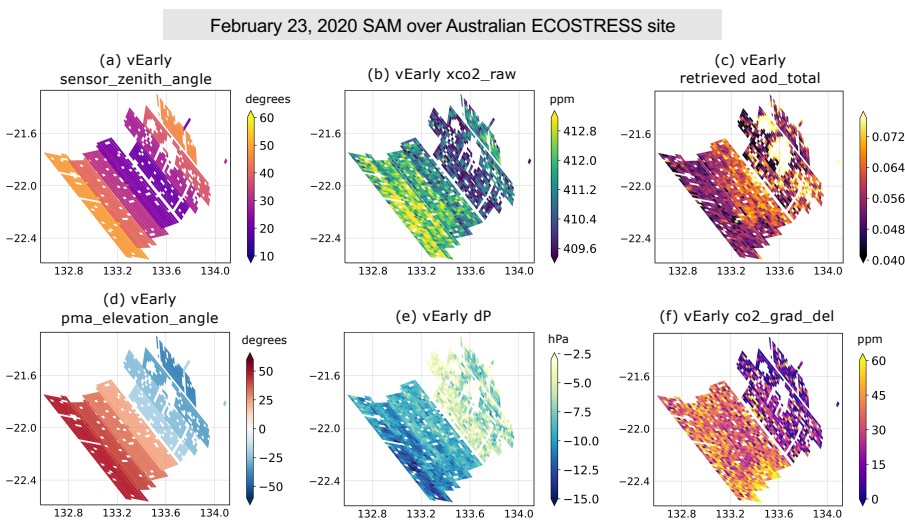

**Figure 6.** Maps of several retrieved and geometry-related parameters for a vEarly SAM over an Australian ECOSTRESS site on February 23, 2020.

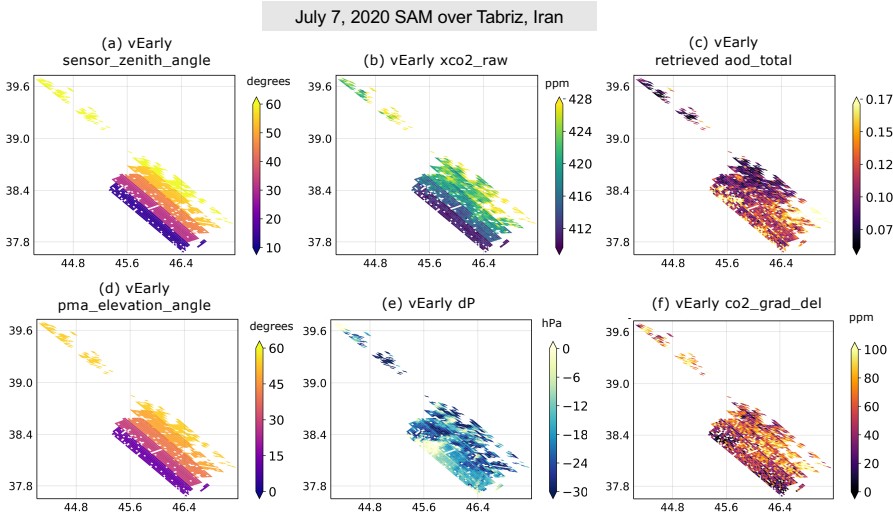

**Figure 7.** Maps of several retrieved and geometry-related parameters for a vEarly SAM over Tabriz, Iran on July 7, 2020.

In panels (c), (e), and (f), we show a few other retrieved parameters which the retrieval might adjust when attempting to characterize geometry-driven aerosol effects. Panel (c) shows the total retrieved AOD (labelled aod_total), and panel (f) shows

330 co2_grad_del, the difference between retrieved and prior $CO_2$ concentration in the upper and lower portions of the retrieved column (the exact calculation can be found in Osterman et al., 2020). The SAM over Iran in Figure 7 may show a hint of SB in these two parameters, but they appear quite smooth in Figure 6. This may be related to higher AODs over the Iran SAM (maximum AOD near 0.35, versus maximum near 0.25 in the Australian SAM, per MODIS estimates) driving stronger geometry-related aerosol scattering effects. Real AOD variations or interactions with heterogeneous surface albedo may dom-

335 inate the visible signal, rather than geometry effects. In panel (d) of these figures, we show dP values, which appear to reflect some SB in Figure 6, but in Figure 7 are dominated by local topography patterns. In these three retrieved parameters - aod_total, dP, and co2_grad_del - we expect that in simulations where geometry is the primary variable, its effects will be visible.

Our first simulations are thus a simple test of this hypothesis. For each date, we apply a constant aerosol profile and type

340 across each scene, set a constant surface elevation and reflectivity, and choose realistic AODs based on local Multi-Angle Implementation of Atmospheric Correction (MAIAC) data, which use MODIS data to derive improved aerosol estimates (Lyapustin et al., 2011; MODIS Land Science Team, 2019, MAIAC data accessed via NASA Worldview). Figures 8 and 9 show the results of these initial simulations - note that the color bar ranges are different than in Figures 6 and 7.

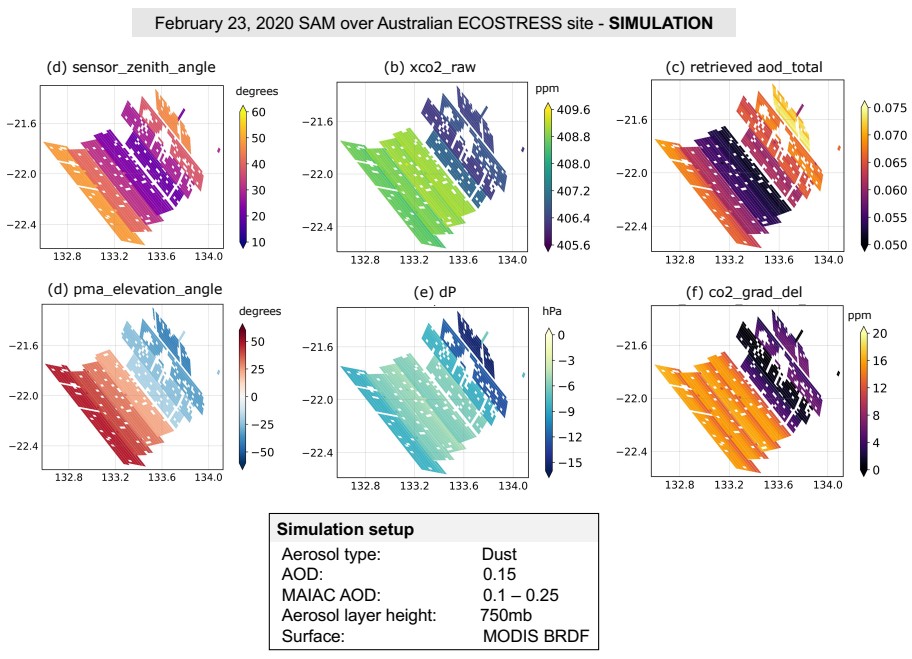

**Figure 8.** Maps of several retrieved parameters from our first simulation of the February 23, 2020 SAM over Australia.

345 We find that simulated spectra derived from simple aerosol scenes are successfully able to generate SB patterns qualitatively similar to those in the operational vEarly data. The general orientation of the observed SB is replicated in both simulated

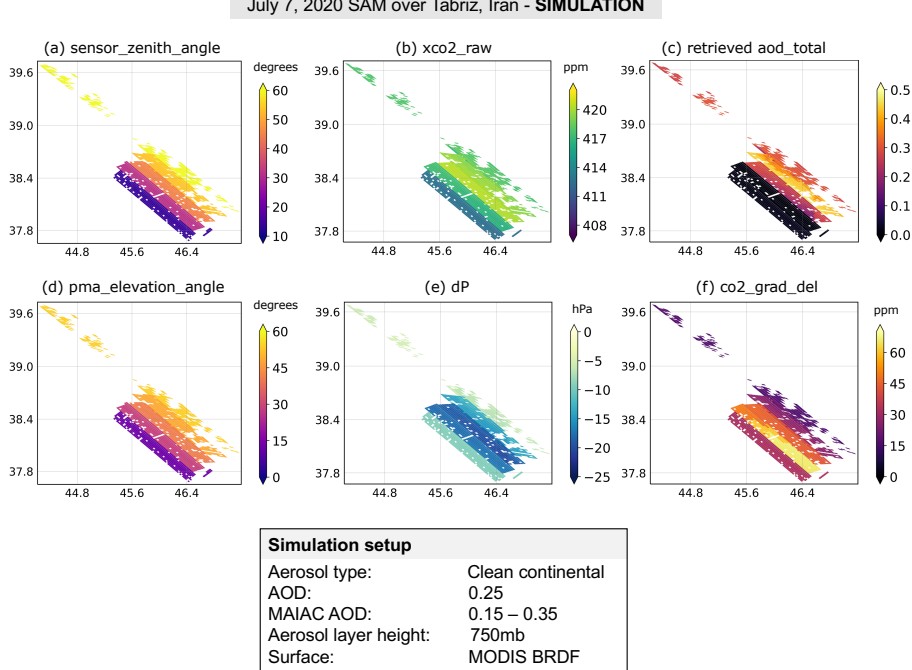

**Figure 9.** Maps of several retrieved parameters from our first simulation of the July 7, 2020 SAM over Tabriz, Iran.

SAMs, but the magnitude of the SB tends to be smaller relative to the observed data. In the absence of more complex aerosol and surface scenes, we do see geometry-driven swath bias in aod_total, co2_grad_del, and dP, although range of values can be quite different than in the observations - for example, the range of aod_total in Figure 9 is much larger than in Figure 7. This may indicate that the aerosol profile constructed for the simulation does not match that observed by the satellite.

We proceed with a series of simulations to evaluate this possibility, testing SB response to various aerosol and surface scenes. Note that because we do not add noise to our simulations, and eliminate many other sources of variability, the denominator of Equation 1 will be small in simulated data compared to the observed data, so the sb_ratio will typically be much larger. We set the sb_ratio = 0.75 threshold for the SB flag using observed data, but it would be larger for simulations - in order to avoid using two different scales of sb_ratio, we will simply use sd_median as a measure of relative SB strength when discussing simulations. We use the terms "sd_median" and "SB strength" interchangeably to describe the same simulated quantity.

## 6.1 Case selections for controlled testing

From the initial simulation tests over Australia and Iran, we single out the Australian non-fossil site as a target for further simulation work. This is a desert site, and should lack influence from local fossil fuel emissions due to its remote location.

**Table 1.** Key metadata and statistics for three Australian ECOSTRESS site SAMs chosen for simulation work

| Date | February 23, 2020 | April 4, 2020 | May 8, 2020 |
|---|---|---|---|
| Orbit # | 4573 | 5204 | 5731 |
| # Soundings | 2104 (2077) | 2465 (2461) | 2337 (2337) |
| Local time | 4:09 PM | 10:10 AM | 11:08 AM |
| Min/max SZA | 55.3/57.0° | 45.6/47.4° | 44.2/44.9° |
| MAIAC AOD | 0.15 | 0.08 | 0.12 |
| sb_flag | 1 (1) | 0 (1) | 0 (0) |
| sb_ratio | 0.98 (1.05) | 0.55 (0.76) | 0.28 (0.28) |
| sd_median | 0.86 (0.76) | 0.37 (0.39) | 0.25 (0.21) |
| mean_sd | 0.88 (0.73) | 0.68 (0.51) | 0.88 (0.75) |

sb_flag, sb_ratio, sd_median, mean_sd are all defined as in Equation 1. These are calculated
for raw vEarly soundings with no filtering applied. Values in parentheses are calculated after
vEarly bias correction and dP_abp filtering are applied. MAIAC AOD is a rough estimate
derived from data available on NASA Worldview.

OCO-3 has taken several SAMs over this site at the time of analysis, at varying times of day and with varying SB signals present according to our criteria. We choose three, which are pictured in Figure 10. Metadata and key statistics are provided in Table 1. By examining three SAMs from the same site, we are able to investigate the differences in atmospheric state and/or observation geometries that drive the operational SB, in addition to using their different geometries as templates for a broader array of synthetic scenes, as mentioned previously.

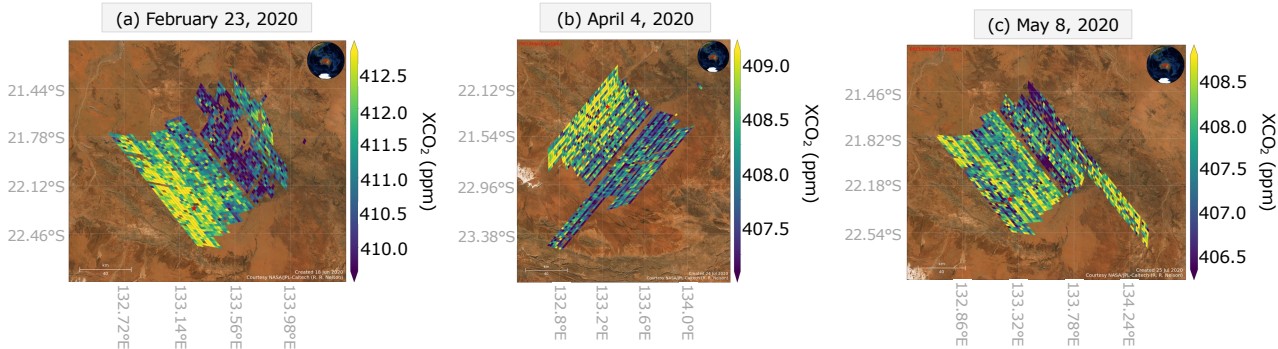

**Figure 10.** Three OCO-3 vEarly SAMs, shown with no filtering or bias correction, all over the same ecostress_au_asm observation site. These three SAMs were chosen for simulation tests due to their varying degrees of apparent SB, as well as different observation and solar geometries.

The strongest example of SB is present on February 23, 2020, which served as our first SAM SB test in the previous section. The sb_ratio (see Equation 1) in this SAM is 0.98 - well over our required threshold of 0.75 for the sb_flag. We also examine

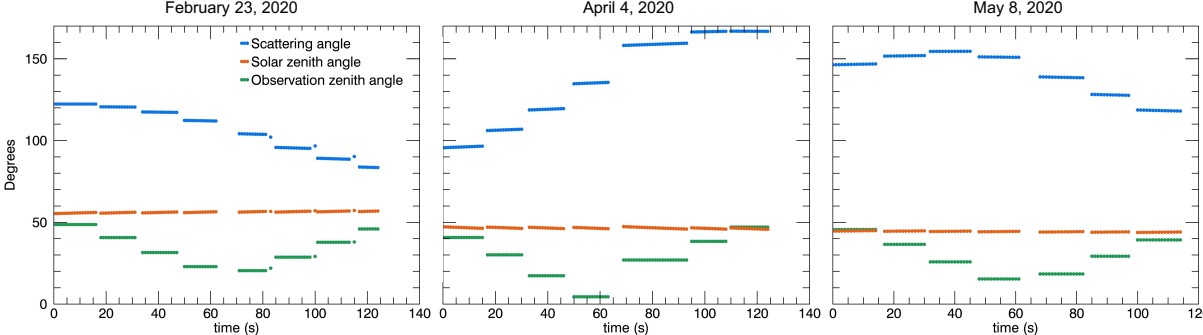

**Figure 11.** Geometry data for the three simulated Australian desert SAMs.

a "borderline" SB case from April 4, 2020, where the raw data has a sb_ratio of 0.55, but after bias correction our sb_flag is triggered by a sb_ratio of 0.76, and some small across-swath gradient of approximately 1 ppm magnitude appears to be present (Figure 10b). The third SAM is from May 8, 2020, and has a very low sb_ratio of 0.28 (Figure 10c). The MAIAC AODs listed in Table 1 will be used in simulations unless otherwise stated. Figure 11 provides context for the observation and solar geome-tries for each of these dates. We include scattering angle, which is the angle between incoming solar photons and outgoing

photons which reach the OCO-3 sensor. Pure forward scattering occurs at 0 degrees and pure back-scattering occurs at 180. The April 4 case has the largest range of scattering angles, approaching 170 degrees in the final two swaths. The February 23 date has the highest solar zenith angle, at 4:09 PM local time, and has the smallest scattering angles.

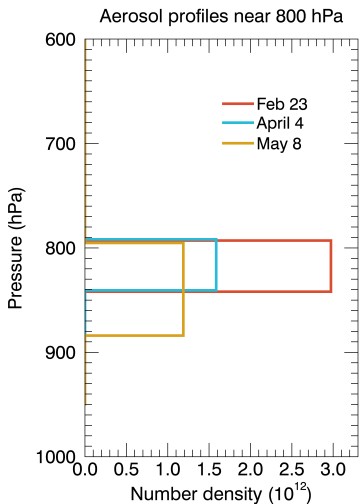

**Figure 12.** Profiles illustrating aerosol layer shape used for the three simulated SAMs. This example shows dust layers whose tops are placed at the native CALIPSO pressure level closest to 800 hPa.

Figure 12 shows the reference aerosol profiles used on each of these three dates. AODs used in this figure match those listed in Table 1, and the tops of the aerosol layers have been adjusted to a height near 800 hPa. In our simulation tests, we will test sensitivity to aerosols by changing the aerosol types (Sect. 6.2), shifting these layers to different vertical levels (Sect. 6.3), scaling the number densities to adjust the AOD (Sect. 6.4). We will further test sensitivity to surface albedo (Sect. 6.5).

## 6.2   Aerosol type testing

For each of the three SAMs, we test two different aerosol types: one coarse mode, a "dust" type, and one fine mode, a "clean continental" type, both of which are realistic given the remote desert setting. The aerosol layers are at a height near 800 hPa, and each date uses the same optical depth (listed in Table 1) for both aerosol types, so only the number densities change.

The results of these tests are shown in Figure 13, and listed in Table 2. We find in the February case that the clean continental aerosol type produces a much stronger SB than the dust mode, with sd_medians of 1.16 ppm and 0.23 ppm, respectively. The May 8 SAM shows a similar result, with stronger SB using the clean continental type aerosol, although the increase in SB is much smaller. The February and May cases have similar AODs - 0.15 and 0.12 - so the smaller SB response on May 8 may be attributed to its smaller mean and range of SZA.

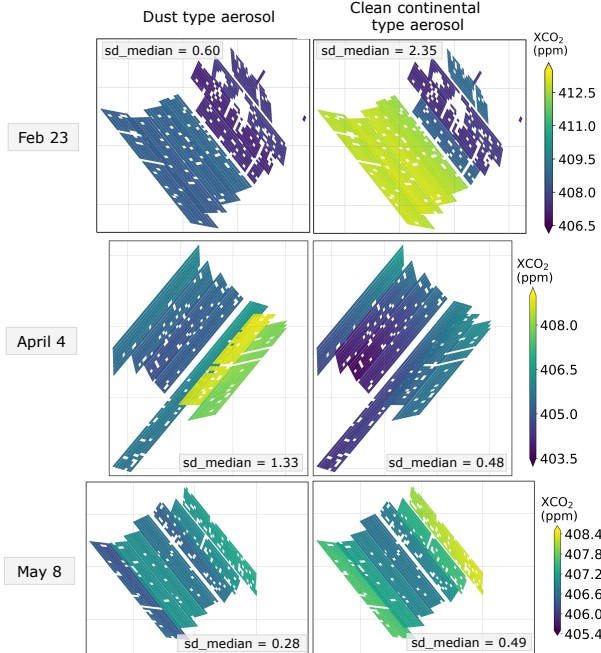

**Figure 13.** Simulated $X_{CO_2}$ using two different aerosol types, for all three SAM dates over the Australian ECOSTRESS site.

The response is opposite for the April 4 case: the dust type aerosol produces a stronger SB. This may be related to the high scattering angle in the final two swaths - nearly 170 degrees, per Figure 11. However, an investigation into the scattering phase

functions for these two aerosol types revealed no correlation relative to the SB strength, for any of the three dates.

**Table 2.** sd_medians (ppm; from Equation 1) for simulated $X_{CO_2}$ in ECOSTRESS AU site SAMs using different aerosol types

| Date | Dust type aerosol | Clean continental type aerosol | Operational |
|---|---|---|---|
| February 23 | 0.60 | 2.35 | 0.86 |
| April 4 | 1.33 | 0.48 | 0.37 |
| May 8 | 0.28 | 0.49 | 0.25 |

Simulations included in this table use realistic AODs for each date based on MODIS/MAIAC estimates, and an aerosol layer lofted to near 800 hPa.

However, in terms of SB orientation, the May 8 case illustrates particularly well the fact that different aerosols produce different geometry-dependent responses: the location of the highest and lowest $X_{CO_2}$ values occurs in different swaths depending on the aerosol type. This makes sense given the unique optical properties of each aerosol type, but would require further study

to predict with quantitative skill. We conclude that the physics of SB, in terms of both magnitude and direction, are highly dependent on the aerosol type, and are complex enough to warrant further study.

### 6.3 Aerosol height testing

Using dust type aerosols, we test a simple profile at three heights for each SAM. We take the base profile for each date, as

shown in Figure 12, and simply loft the aerosol layer to the desired pressure level: either 900 hPa (near-surface), 800 hPa (mid-level), or 750 hPa (higher-level). These are chosen arbitrarily to represent a typical range of tropospheric aerosol heights, spanning roughly a few vertical kilometers.

We also include a clear-sky, no-aerosol run as a baseline for retrieval behavior. Figure 14 shows the results of this experiment using a dust type aerosol with AOD of 0.15 in the February 23, 2020 case. Table 3 shows that there is a positive correlation

between aerosol height and sd_median - i.e., as the aerosol layer is lofted, the strength of the SB increases.

As indicated by Table 3, the April 4 and May 8 cases respond similarly to changes in aerosol layer height. The SB is always stronger in simulations with aerosol than in simulations without; it increases in strength as we move the aerosol layers higher in the column. The relative increase between the 900 hPa and 750 hPa aerosol layer varies between cases - the April 4 case has the strongest change in SB, which again could be related to its broad range of scattering angles, and high scattering angles near

the end of the scan. The range of sd_medians on May 8 is notably smaller than that of either February 23 or April 4, similar to its response to different aerosol types. This could again be due to May 8 having the lowest range of SZAs relative to the other two dates, per Table 1 and Figure 11.

The correlation between SB strength and aerosol layer height is consistent for both large and small aerosol types, as re-

vealed by additional testing using the clean continental aerosol type. Operationally, this makes SB more likely to be detectable

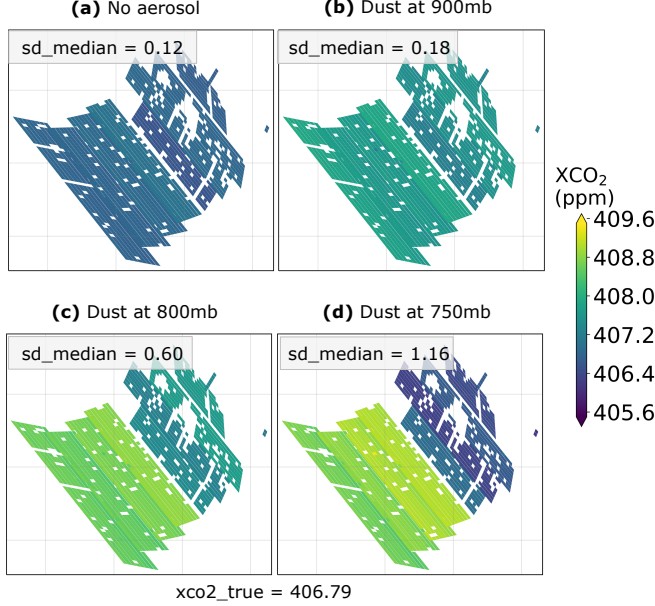

**Figure 14.** Simulated $X_{CO_2}$ using a dust type aerosol at three different heights, with an AOD of 0.15, over the Australian ECOSTRESS SAM site on February 23, 2020.

**Table 3.** sd_medians (ppm) for simulated $X_{CO_2}$ in ECOSTRESS AU site SAMs using different aerosol heights

| Date | No aerosol | Aerosol near 900 hPa | Aerosol near 800 hPa | Aerosol near 750 hPa | Operational |
|------|------------|----------------------|----------------------|----------------------|-------------|
| February 23 | 0.12 | 0.18 | 0.60 | 1.16 | 0.86 |
| April 4 | 0.22 | 0.47 | 1.33 | 1.81 | 0.37 |
| May 8 | 0.18 | 0.21 | 0.28 | 0.39 | 0.25 |

Simulations included in this table use realistic AODs for each date based on MODIS/MAIAC estimates. All use a dust type aerosol.

beyond scene-typical noise, both by eye and by our SB flag, if there is aerosol higher in the atmospheric column. In addition to differences in viewing geometry, this could be one reason that SB appears in some cases and not others in real measurements.

## 6.4 Aerosol optical depth testing

In this section we test three aerosol optical depths for each SAM: one low (0.1), one moderate (0.2), and one high (0.35). These are chosen to represent a realistic range typically seen over the Australian ECOSTRESS site according to MAIAC. We use a single layer of dust aerosol at a height near 800 hPa. Per Table 4, we find that SB strength increases with increasing AOD

in all three Australian SAMs, with April 4 as the exceptionally strong SB case once again - the $X_{CO_2}$ values change much more significantly in the final two swaths with high scattering angles than in the others. As in the aerosol height tests, the SB is always stronger in the presence of aerosols than in the AOD =0.0 test case. We are therefore more likely to see SB over high-AOD scenes.

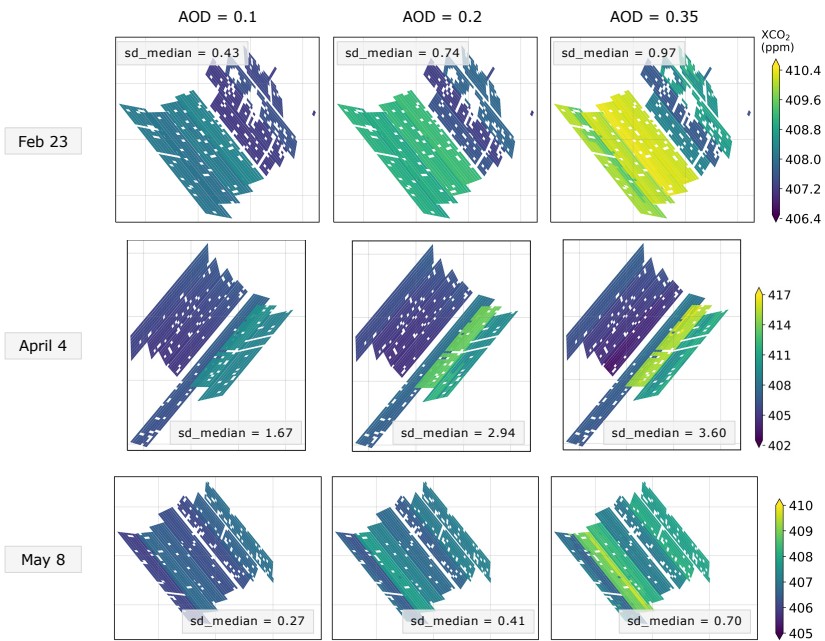

**Figure 15.** Retrieved $X_{CO_2}$ using a simulated dust type aerosol near 800 hPa and two different optical depths, for all three simulated SAMs.

**Table 4.** sd_medians (ppm) for three different AODs in each simulated SAM

| Date | AOD = 0.0 | AOD = 0.10 | AOD = 0.20 | AOD = 0.35 |
|---|---|---|---|---|
| February 23 | 0.12 | 0.43 | 0.74 | 0.97 |
| April 4 | 0.22 | 1.67 | 2.94 | 3.60 |
| May 8 | 0.18 | 0.27 | 0.41 | 0.70 |

Simulations included in this table use realistic AODs for each date based on MODIS/MAIAC estimates. All use a dust type aerosol.

## 6.5 Surface albedo testing

Per discussion in Sect. 3, a larger fraction of scenes with high retrieved surface albedo, in any of the three spectral bands, are flagged as having SB. In this section, we assign a range of surface albedos to our simulated SAMs to strategically test the SB

response.

We test surface albedos ranging from 0.1 to 0.6 in each of the three bands, which covers a realistic range of albedos over land. When varying one band, we hold albedo constant in the other two bands at a realistic value for this site: $O_2$ A-band (O2A) at 0.25, strong $CO_2$ band at 0.25, and weak $CO_2$ band at 0.30. The functionality of the simulator requires that when prescribing a specific surface albedo, we model the scene using a Lambertian surface. (This is different from the previous sections, where the simulator always utilizes a BRDF surface.) The ACOS retrieval, however, uses a BRDF surface model.

These simulations use the same vertical aerosol distributions as shown in the previous sections; the AOD on each date is the realistic MAIAC value listed in Table 1, and each layer is near 800 hPa. All three dates use the dust aerosol type.

The top row of Figure 16 shows the results of the albedo tests for all three simulated SAM dates, with no filtering applied to the simulated retrievals. On each date, we compare the same number of soundings across all six albedos. The SB is highest at lower albedos, and particularly for the February and April SAMs; this is the opposite trend we expect based on our analysis in Sect. 3. However, over low O2A albedos, our dP_abp filter removes some soundings that contribute to swath bias, as shown in the bottom row of Figure 16. The dP_abp filter does not, however, remove any soundings when the WCO2 and SCO2 albedos are low. This makes some sense, as the $O_2$ A-band is most skilled at seeing aerosols, and would best pick up on them with a high contrast between a dark, low-reflective surface and a bright, high-reflective aerosol.

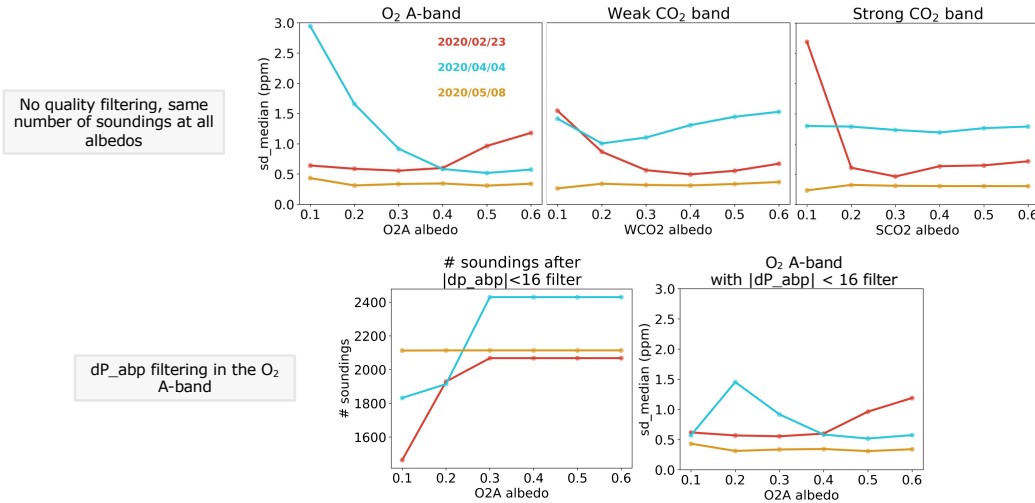

**Figure 16.** SB strength (sd_median) depending on surface albedo in each of the three OCO-3 bands. The top row includes no quality filtering, and the number of soundings per day is the same for each albedo run. The bottom row shows how dP_abp filtering affects sd_medians the $O_2$ A-band (it has no effect in the other two bands).

From this exercise, we surmise that the trend toward SB at higher albedos observed in Sect. 3, Figure 5 is not because SB is more likely to occur over bright scenes. In fact, stronger SB tends to occur at lower albedos. The retrieval, however, is more skilled at differentiating aerosols over dark surfaces, whereas it has trouble identifying aerosols over bright surfaces. This makes our quality filtering more effective at removing aerosol-related effects - including those of SB - over dark surfaces, so that SB is more likely to slip past our filters when the surface is bright. This conclusion reinforces the fact that the presence of

SB is intimately linked to the presence, and our retrieval's treatment, of aerosols.

## 6.6    Simulation Summary

In our simulation work, we show that the large changes in $X_{CO_2}$ between swaths are primarily correlated to the changing viewing geometry, and that by eliminating other sources of variability within a SAM, we can simulate the purely geometry-driven response within the retrieval. We choose three SAMs over the same Australian desert site which represent a range of

SB signals, solar zenith angles, and scattering angles. For each SAM, we test an aerosol-free scene, and four parameters: aerosol type, aerosol height, aerosol optical depth, and surface albedo. Aerosol-free scenes suffer little to no SB, but we find that SB increases with both aerosol height and optical depth, and that the aerosol type changes the SB strength in different ways depending on some complex interplay of aerosol optical properties and geometry. Lower surface albedos tend to induce a stronger SB, again depending on the observation and viewing geometry - but quality filtering using the $O_2$ A-band is skilled

at identifying aerosols over dark surfaces and effectively removes erroneous measurements, leaving more SB at higher albedos where it is harder to distinguish aerosols from the bright surface beneath.

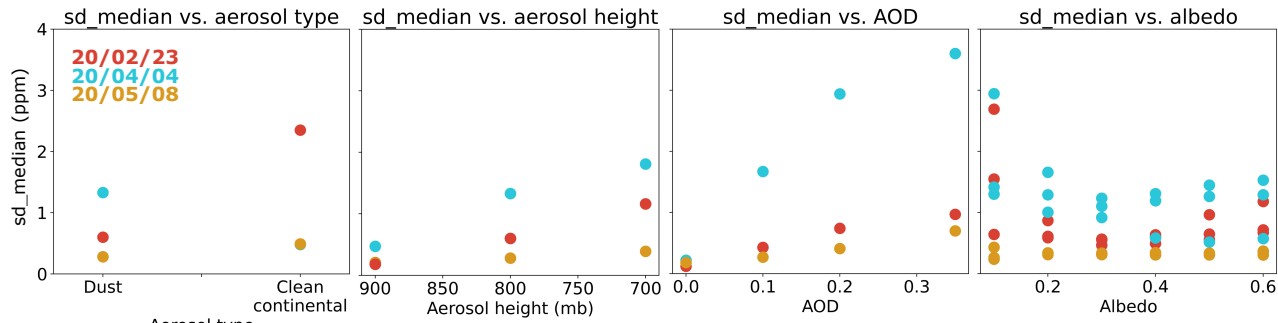

**Figure 17.** sd_median versus each of our tested parameters.

Figure 17 shows sd_median versus each of our independent parameters on the same scale. We attempt to represent some realistic range of values of each parameter, in order to be able to compare their effects. All four parameters have the potential to induce quite a large SB in our simulated SAMs, with high AODs inducing the highest sd_medians of more than 3 ppm.

We only tested AODs as high as 0.35, although higher AODs are sometimes observed - but typically, those are eliminated by quality filtering. Low O2A albedos produced the second strongest SB in our simulations, but similarly are often mitigated by our dP_abp filter. We acknowledge that none of the sd_medians shown here are as high as are sometimes observed in other

SAMs - occasionally exceeding 10 ppm, such as in the Tabriz example shown in Figure 7 - but we hypothesize that testing an even broader range of aerosol types, optical depths, or observation angles would produce such results. Biases or real variations in $X_{CO_2}$ derived from additional complexity in real scenes can also enhance this signal - varying surface albedo or topography, for instance.

These simulation studies reveal that SB is primarily and intimately connected to the presence of aerosols and the interplay of their optical properties with the solar and instrument viewing geometries. We now have a better understanding of the types of scenes that are likely to suffer SB - those with high aerosol depths, or aerosols that are lofted higher in the atmospheric column, and in geometries with broader ranges of observation and solar zenith angles. Future work may involve a more detailed study of how the physics of aerosol optical properties and viewing/solar geometries combine to produce a SB response.

## 7 Swath bias in OCO-3 version 10

Our analysis thus far has focused on vEarly, but the recent OCO-3 version 10 (v10) dataset includes a number of updates which have the potential to mitigate SB effects. This section provides a brief summary of those changes, and explores their effect on SB in the v10 SAM collection.

Key v10 updates relevant to our study can be summarized as follows:

– A geolocation error of up to 20 kilometers in vEarly was reduced, with more current values ranging a few hundred meters to about 2 km.

– Improved calibration algorithms account for lamp aging.

– A longer empirical orthogonal function (EOF) training dataset for v10 captures changes in spectral shape due to ice buildup on focal plane arrays.

– Sounding selection criteria for v10 L2FP processing were updated to include the IDP $CO_2$ and $H_2O$ ratios, which significantly reduces the number of soundings processed in certain v10 SAMs relative to vEarly.

– Operational L2FP post-processing quality flags in vEarly eliminated a progressively larger fraction of "good" soundings. In order to see SB effects, we instead use a dP_abp filter in this work. Updated quality flags in version 10 retain a lower, more realistic number of soundings per SAM compared to vEarly operational flags, and eliminate a larger number of soundings relative to our dP_abp filter. This change is highly coupled with the EOF improvements described in the previous bullet.

– A parametric $X_{CO_2}$ bias correction term was added in v10 accounting for biases related to the retrieved weak $CO_2$ band albedo, derived from a training dataset which covers a longer time period than that of vEarly.

- – Based on the larger training dataset, all coefficients in the parametric bias correction were updated. This includes the dP
bias correction term coefficient, which was increased from smaller than -0.1 to -0.62. The new value is more in line with
expectations, based on experience with OCO-2.

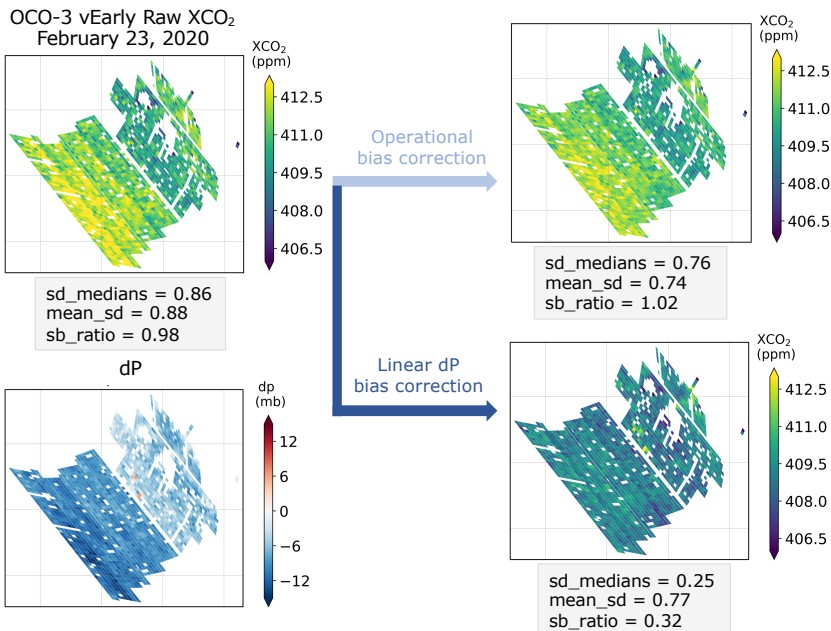

**Figure 18.** February 23, 2020 vEarly SAM from OCO-3, showing the effects of the operational bias correction and a linear dP-derived bias correction on the observed SB. The linear dP bias correction is more similar to the bias correction in v10.

Operational v10 quality filtering is much more thorough than our simple dP_abp attempt, and as far as bias correction, from our work with vEarly, we expect that the stronger dP bias correction term in particular may have a significant effect on SAMs' SB. In particular, our February SAM exhibited some correlation between $X_{CO_2}$ and dP, and we see a similar pattern in a number
of other vEarly SAMs. For mitigation of SB, simple linear bias correction derived solely from dP proves more effective in those cases than the vEarly operational bias correction. Figure 18 shows this phenomenon for the February 23 vEarly data. This may make some sense relative to our simulation results, which shows that aerosol properties have the most influence on the SB: the retrieval might account for complex aerosol effects by over-correcting surface pressure. Some, but not all, of our simulation configurations did show a similar correlation between dP and $X_{CO_2}$. Thus the improved dP correction seems likely to improve
SB, via mitigation of aerosol effects, in some scenes but not all. The dP bias correction in v10 will be more effective generally than in vEarly, because the dP correction assumes accurate prior surface pressures, which is not the case in the presence of the significant geolocation errors suffered in vEarly.

For our comparison between v10 and vEarly, we evaluate a total of 5459 SAMs from v10 (specifically, version B10306r) which have vEarly counterparts. These SAMs span August 2019 to June 2021. For the most direct comparison, we first apply
the more restrictive v10 sounding selection criteria to vEarly. We apply quality filtering and bias correction, narrow down to

only SAMs with at least 500 soundings ("N>500" representing the number of soundings, "NSAM" representing the number of N>500 SAMs), and calculate our SB parameters from Equation 1. vEarly quality filtering is our custom | dP_abp | < 16 hPa filter, and v10 is filtered using the operational v10 quality flags. Figure 19 details the comparison.

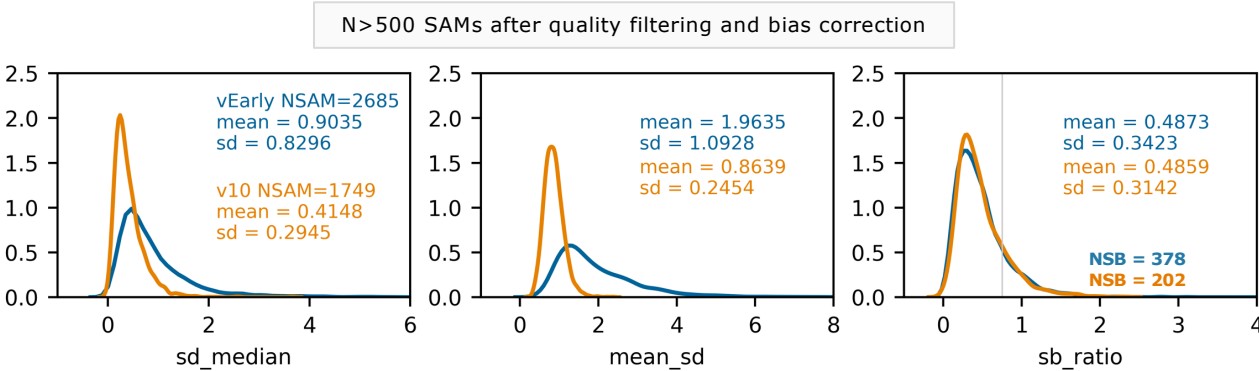

**Figure 19.** Normalized histograms of swath bias parameters for vEarly and v10 SAMs, using v10 sounding selection for both datasets. All SAMs have at least 500 soundings (N>500). The number of SAMs in each dataset is given by NSAM. vEarly is in blue, v10 is in orange, and the SB threshold of sb_ratio > 0.75 is indicated by the gray line. The number of swath bias cases is listed as NSB in the right-hand panel.

Recall from Sect. 3 that sb_ratio is the result of dividing sd_median by mean_sd, where sd_median is essentially a measure of swath-to-swath bias and mean_sd a measure of within-swath - or within-scene - noise. Figure 19 shows that the swath-to-swath bias is greatly decreased with filtering and bias correction in v10; the same is true of the noise within-swath noise. We conclude that swath bias in v10 looks distinctly different than in vEarly: the magnitude is much smaller, and exists within a much smoother $X_{CO_2}$ field. The distribution of the sb_ratio does look similar to vEarly, but there are fewer SAMs with SB in

v10. Quality filtering in v10 decreases the number of N>500 SAMs fairly significantly, which we believe is a more realistic result than the more permissive dP_abp filtering used for vEarly. Per this exercise, out of the full 5459 SAMs compared, vEarly contains 6.9% SB cases, compared to only 3.7% in v10.

Quality filtering and bias correction each have their own effect on the swath bias, and improvements due specifically to the v10 bias correction could be derived either from the improved dP correction or the new weak $CO_2$ albedo correction, although

we suspect the dP term as previously discussed. To separate these effects, we apply each of the v10 parametric bias correction terms to the raw $X_{CO_2}$ data individually, and evaluate the change in SB strength (defined as the sb_ratio from Equation 1). We perform this test for a subset of 475 cases with sb_flag = 1 in the raw v10 data. Figure 20 shows the results of this exercise.

The dP correction indeed has the largest impact in reducing the SB, in line with our hypotheses regarding both aerosol

effects and geolocation. The median SAM's SB is weakened by 20% with this bias correction term applied. The weak $CO_2$ albedo correction has a much smaller effect on the SB overall, with a median SB change of -1%. Other bias correction terms

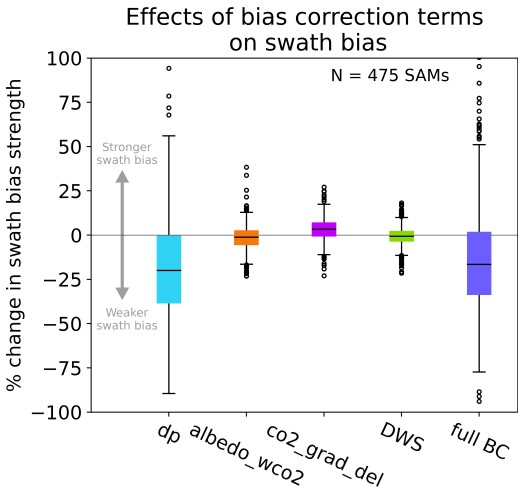

**Figure 20.** SB response to individual v10 bias correction terms, for 475 v10 SAMs whose sb_flag = 1 prior to filtering and bias correction. This set of SAMs extends beyond the time period used in the vEarly comparison.

have a similarly minor effect on the SB compared to dP, and the full bias correction with all terms applied reduces SB in most SAMs, with a median SB change of -16.5%. We do note, however, that in tests applying the bias correction and quality filtering separately, the quality filtering had the more substantial effect on the SB: bias correction alone reduced the number of v10 SB SAMs from 325 (in raw data) to 310, and quality filtering reduced it from 325 to 225. While dP had the largest impact within the bias correction, the quality filtering had an even larger impact, indicating that the swath bias is not driven specifically by dP, but rather by extreme aerosol effects being characterized poorly within the retrieval.

All three of the Australian ECOSTRESS SAMs used in our simulation work benefit from v10 improvements. Figure 21 shows a comparison between their vEarly and v10 bias-corrected and quality-filtered counterparts. None of these cases are flagged as having SB in v10, all with sb_ratios less than 0.75. Between the three, several v10 improvements are apparent.

In the February 23 SAM, geolocation and calibration improvements remove the across-swath gradient almost entirely, and improved quality flags also remove many soundings where we retrieve higher aerosol optical depths. On April 4, improved geolocation and bias correction sharpen a few thin topographical features stretching northwest to southeast, indicating a topography effect such as those described in (Kiel et al., 2019). A footprint-dependent bias is also removed by bias correction in v10 - we see the same bias improved in the May 8 SAM. On May 8 the v10 bias correction also removes a signal in the northeast corner of the SAM, which we believe is related to a real aerosol signal manifesting in dP, per MAIAC data over the site. The sb_ratio is reduced in v10 for all three SAMs.

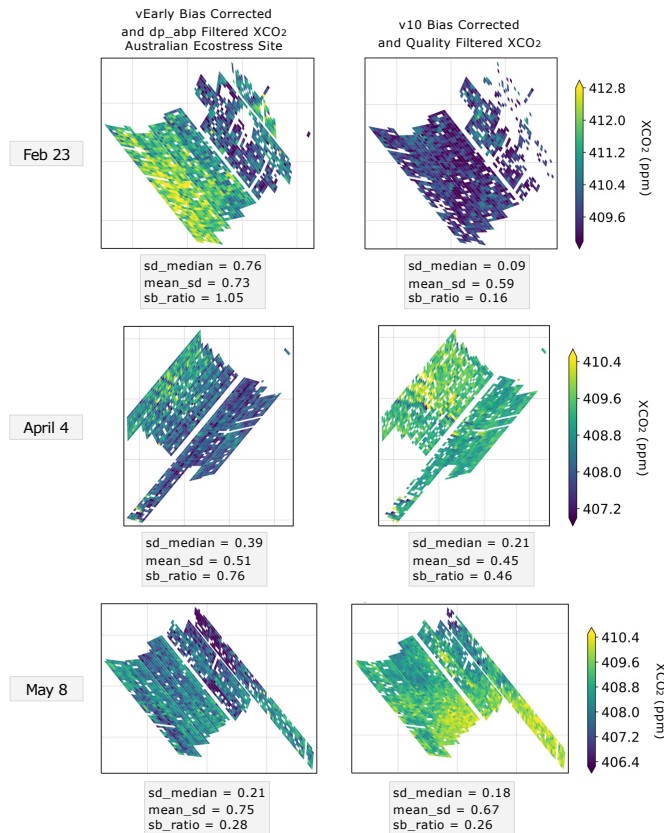

**Figure 21.** The three Australian SAMs used in our simulation work, comparing the operational products from vEarly (left) and v10 (right).

Aside from our three Australian SAMs, we perform some visual analysis of 243 quality-filtered, bias-corrected SB SAMs from the v10 dataset extending through October 2021. In an attempt to estimate how many of these SAMs seem to suffer a genuine geometry-related SB and how many simply contain across-swath $X_{CO_2}$ gradients derived from other sources, we look at maps of raw $X_{CO_2}$, bias-corrected and quality-filtered $X_{CO_2}$, dP, retrieved albedo, and TROPOMI $NO_2$ (Veefkind et al., 2012; Van Geffen et al., 2019). We estimate that at least half of the 243 SAMs are flagged as SB cases due to $X_{CO_2}$ variability caused not by geometry/aerosol effects, but by other biases (e.g., topography, albedo); real $X_{CO_2}$ signals, including some with fossil signals which are verifiable using TROPOMI $NO_2$ data; or other unknown sources. An example of fossil emissions triggering a false SB flag is shown in Figure 22, on the coast of Saudi Arabia. TROPOMI $NO_2$ indicates a real $X_{CO_2}$ plume, but despite the v10 quality flags removing a generous number of soundings, enough of the OCO-3 SAM remains after bias correction and filtering to see an across-swath gradient which is derived from the contrast of the fossil fuel signal and background. Figure 23 shows an example of a v10 SAM with true swath bias: both the raw and bias-corrected data show a shift of nearly 1 ppm in $X_{CO_2}$ between swaths, which may drown out any existing fossil signal as observed by TROPOMI $NO_2$. The swath bias

appears to be due to higher aerosol loading in the scene, with MAIAC indicating AODs of nearly 0.3.

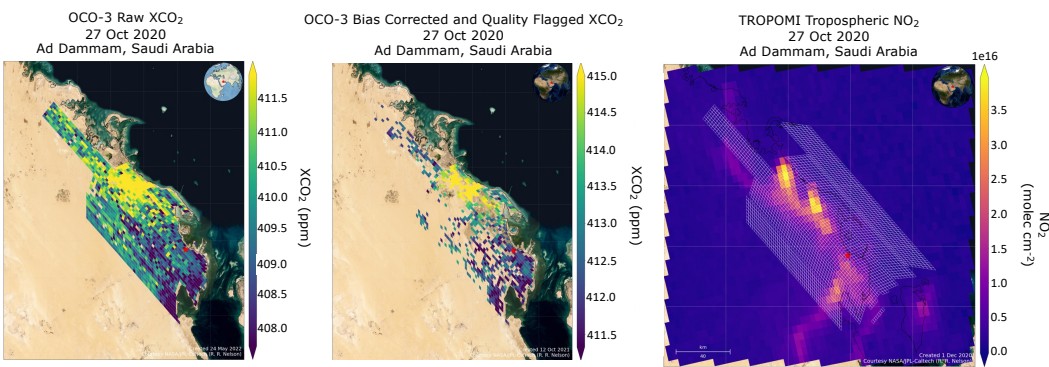

**Figure 22.** A v10 OCO-3 SAM whose SB flag is triggered by a real across-swath $X_{CO_2}$ gradient, due to a fossil fuel emission signal clustered on one side of the SAM, rather than a geometry-related bias.

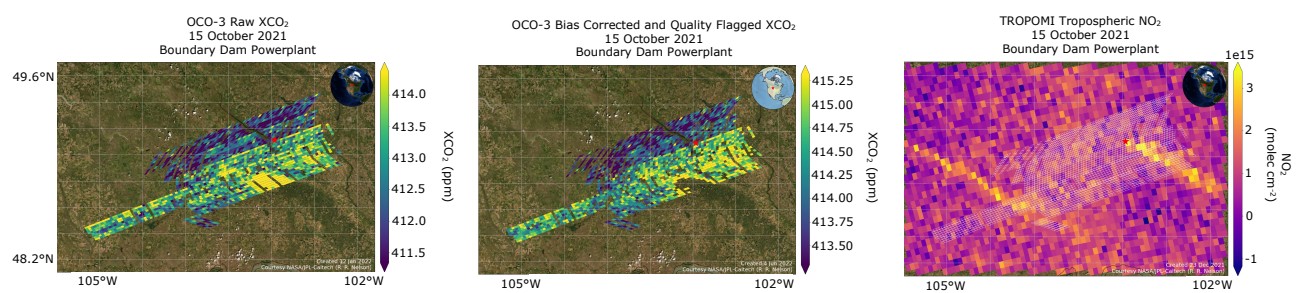

**Figure 23.** A v10 OCO-3 SAM at Boundary Dam power plant in Saskatchewan, Canada which appears to have a true swath bias, even after bias correction and quality filtering. The SAM was taken at 16:42 UTC (10:42 AM local time), and MODIS data indicate high AODs of around 0.3 in the area on this date.

We acknowledge that vEarly suffers the same limitations as v10 in terms of this interpretation of the sb_flag, indicating that vEarly likely also suffers a far smaller number of "true" SB cases than our flag suggests. Despite this, we still consider v10 an improvement based on the across-swath and within-swath improvements to the $X_{CO_2}$ field shown in Figure 19.

There are a small number of SB-affected SAMs in v10 which are not flagged as such in vEarly, which we find is due to a less noisy $X_{CO_2}$ field overall - that is, a reduction in the denominator of Equation 1. We don't believe most of these cases suffer from significant SB. Overall, SB appears to be driven primarily by aerosol effects, and as shown in Sect. 6, our retrieval generates its own SB in certain instances. However, v10 quality filters prove especially effective at mitigating swath biases, and the dP term in the v10 bias correction appears to remove some SB even after quality filtering, because of the way aerosol effects are often folded into dP. Therefore, while it still exists, filtering and bias correction largely appear to mitigate SB in

OCO-3 v10 data.

## 8   Conclusions

Target mode data from the OCO-2 and OCO-3 missions, and SAM data from OCO-3, provide us with an avenue to explore not only local-scale $CO_2$ emissions, but also geometry-related biases in spaced-based $CO_2$ measurements, since they sample a nearly-instantaneous atmosphere from a range of viewing angles. With the arrival of the first operational OCO-3 SAM data, we observe a new geometry-related bias, in the form of a swath-dependent $X_{CO_2}$ gradient spanning several parts per million. We refer to this as "swath bias."

In this study we develop a set of criteria to detect swath bias (SB) in any given SAM, by calculating the ratio of swath-to-swath scatter in the $X_{CO_2}$ field to the scatter over the full scene and triggering a swath bias flag (sb_flag) over a threshold value of 0.75 (Equation 1). Per these criteria, we find that roughly 12% of SAMs in the vEarly dataset suffer from SB, and 256 of those 352 cases are over urban/fossil sites. An analysis of key retrieval parameters reveals a relationship between the SB and local time of day, correlated to solar zenith angle. Together these describe a correlation to path length, which could be related to viewing and solar geometry, as well as aerosol optical depth - and we do observe a higher fraction of SB at SAMs with higher retrieved optical depths. There is also a higher frequency of SAMs over scenes with higher retrieved albedo. We turn to the L1b simulator framework to study these relationships in a more systematic manner.

By studying real OCO-3 SAMs, we show that SB in $X_{CO_2}$ is primarily correlated to the viewing geometry, though the geometry signal in other retrieved parameters appears easily obscured by signals from other sources, such as heterogeneous aerosol fields or surface topography. To remove additional in-scene variability, we build custom aerosol and surface scenes to generate simulated radiance spectra, and use those spectra to retrieve $X_{CO_2}$ with the ACOS L2FP algorithm. Our first tests show that we can successfully reproduce geometry-driven SB effects via this method, and while not identical to the observed vEarly SB, the similarities are enough to instill confidence in the utility of the procedure. We select a set of three SAMs over an Australian desert site, which represent a range of solar zenith angles and viewing geometries, as well as a range of SB effects. For these three SAMs, we perform a series of controlled tests, changing individual inputs to the simulated scene.

Scenes with no aerosol show little to no SB, but each SAM's response to aerosol height and optical depth is consistent - the higher the aerosol within the column, or the higher the AOD, the stronger the SB. We test one coarse and one fine mode aerosol for each SAM, and each produces a different SB, but which is stronger depends on the conditions of the SAM. The precise combination of geometry and aerosol optical properties required for a strong SB appears to be complex, although it is apparent that the unique optical properties of each aerosol type interact with the geometry to produce the different $X_{CO_2}$ patterns. The

exact nature of these interactions would require further study to describe in a quantitative manner.

Our manipulation of surface albedo, testing a range of values from 0.1 to 0.6 in each of the three OCO-3 bands, reveals that swath bias is more likely to occur over dark surfaces with low albedos. Our $O_2$ A-band filter is skilled at mitigating the effect over dark surfaces, however, where aerosols are differentiable from the surface beneath. It has more trouble discerning aerosols and their effects over bright surfaces, and is less likely to filter out affected soundings. We believe this behavior may account for the observed trend of more SB SAMs at higher albedos in vEarly, and reinforces the fact that SB is intimately linked to aerosols.

Finally, we replicate our vEarly analysis using the updated version 10 dataset, and see vastly improved statistics. We find that improved quality filtering is the primary driver of this development, removing a significant number of soundings in SB-affected SAMs. Better sounding selection and significant geolocation improvements, combined with a better dP bias correction, also make v10 more effective at mitigating SB effects, and generally improve the quality of the final data product. We estimate that at least half of the remaining v10 SB SAMs, after filtering and bias correction, may in fact be triggered by other biases or real $X_{CO_2}$ signals, but further investigation is required to quantify these numbers accurately. For the remaining truly geometry-related SB, we hypothesize that retrieval algorithm adjustments, with increased complexity or better prior aerosol information, may help mitigate the SB signal in the final Level 2 product. They may even be due to smaller remaining OCO-3 geolocation errors. Additionally, alternate or supplemental post-processing may prove effective. We leave these approaches to future study.

We conclude that the swath bias effect is intrinsically linked to the presence of aerosols, their interplay with observation geometry, and our ability to filter out aerosol-affected soundings, as well as the way aerosols are characterized within our retrieval. There has long been room for improvement in the latter, and it is a critical piece of the retrieval puzzle with further complexity to be studied. Work by Rusli et al. (2021), in support of the European Space Agency's $CO_2$ Monitoring (CO2M) mission (Ciais et al., 2017; Janssens-Maenhout et al., 2020), showed with synthetic data that jointly retrieving $X_{CO_2}$ and aerosol information provided by a multi-angle polarimeter (MAP) can significantly improve aerosol-related biases, overall bias, and spread in the resulting $X_{CO_2}$. Strategies along these lines - complimentary, aerosol-dedicated instruments alongside spectrometers measuring trace gases - may be one effective approach to improving the biases we see in this study. Accurately representing aerosols in greenhouse gas retrievals will prove just as important to future $CO_2$-monitoring missions as it has to OCO-3 - perhaps even more so as the remote sensing community continues to hone in on local-scale emissions.

*Code and data availability.* OCO-3 data products are available at https://disc.gsfc.nasa.gov/datasets?keywords=OCO-3&page=1. Carbon-Tracker CT2019B is available through NOAA GML at https://gml.noaa.gov/aftp/products/carbontracker/co2/CT2019B/molefractions/co2_total/. NCEP meteorological data are found at https://psl.noaa.gov/data/gridded/data.ncep.reanalysis.html, and the MODIS MCD43A1 prod-

uct is at https://lpdaac.usgs.gov/products/mcd43a1v006/. The left panel and insets of Figure 3 were created using visualization software developed at CSU/CIRA, the code for which is available at https://github.com/hcronk/oco_vistool.

*Author contributions.* E. Bell completed the bulk of the analysis and wrote the initial draft of the published work. C.W. O'Dell further advised on retrieval behavior, scientific direction, methodology, and acquired funding for this work. T. Taylor served as mentor on the software side, and contributed data, retrieval, and scientific guidance. A. Merrelli also provided key input on retrieval algorithm behavior and scientific analysis. R.R. Nelson assisted with case selection, lent data- and retrieval-related expertise, and contributed a myriad of figures. M. Kiel provided data and retrieval insights; A. Eldering guided the focus of this work, along with that of the OCO-3 mission; and R. Rosenberg and B. Fisher were (and are) crucial to the development and record of the latest and greatest OCO-3 data products. All authors contributed to the editing of the paper.

*Competing interests.* We declare no competing interests in this work.

*Acknowledgements.* The authors would like to thank the OCO-3 science team at large for the making the most of such an impactful, yet relatively brief, mission. Additional thanks to Greg McGarragh, Peter Somkuti, and Heather Cronk for their assistance with retrieval infrastructure, software questions, and data visualization. This work was funded thanks to NASA ROSES Grant 80NSSC21K1078 and NASA JPL subcontract 1557985.

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
