# Peer review of "Exploring bias in OCO-3 Snapshot Area Mapping mode via geometry, surface, and aerosol effects"

_Atmospheric Measurement Techniques, 2022_

## Author Response (AR1)

We thank the reviewers for their time and insightful comments, and hope that we have addressed them satisfactorily! Reviewer 2 responses begin on Page 13.

- https://doi.org/10.5194/amt-2022-241-RC1

REVIEWER 1

I am missing some relevant references which I recommend to add (see also below). For example, I recommend to add Reuter et al., 2019, when citing Nassar et al., 2017, as this is another study where OCO-2 data have been used to obtain information on power plant CO2 emissions. I also recommend to add references to the future missions listed in the paper (MicroCARB, GeoCarb, GOSAT-GW, CO2M). In particular, Rusli et al., 2021, should be cited as their investigation on aerosol related XCO2 biases is relevant for this publication, which also highlights aerosol related issues.

- Reuter et al. (2019) reference has been added to line 34:
  "[...] on scales as fine as those of individual power plant plumes (Nassar et al., 2017; Reuter et al., 2019). "
- and to line 62-63:
  "[...] Indeed, co-located $NO_2$ observations have been shown to be helpful in plume identification when using OCO-2 data Reuter et al. (2019)."
- as well as described in conjunction with Nassar et al. (2017, 2021) and Kiel et al. (2021) in lines 144-146:
  "Reuter et al., 2019 similarly showed the value of combining OCO-2 observations of power plant plumes with $NO_2$ observations from the Sentinel-5 Precursor over six sites, comparing their estimates of cross-sectional fluxes to existing emission inventories successfully within their uncertainties."
- Rusli et al. (2021) is now mentioned in the Conclusions section, in lines 642-45:
  "Work by Rusli et al. (2021), in support of the European Space Agency's CO2 Monitoring (CO2M) mission (Ciais et al., 2017; Janssens-Maenhout et al., 2020), showed with synthetic data that jointly retrieving $X_{CO2}$ and aerosol information provided by a multi-angle polarimeter (MAP) can significantly improve aerosol-related biases, overall bias, and spread in the resulting $X_{CO2}$ ."
- Lines 87-90: "Future missions with similar GHG-monitoring strategies, such as MicroCARB (Pasternak et al., 2017; Bertaux et al., 2020), GeoCarb (Moore III et al., 2018; Nivitanont et al., 2019), GOSAT-GW (Kasahara et al., 2020), and CO2M (Ciais et al., 2017; Janssens-Maenhout et al., 2020), may benefit from an improved understanding of these types of biases."

**Specific comments:**

**Q1:** Page 2, line 25 following: Sentences "Since the launch of the Greenhouse gases Observing Satellite (GOSAT; Kuze et al., 2009; Yokota et al., 2009) in 2009, space-based instruments have been addressing the particular challenge of scale. In decades prior, the global carbon cycle was studied using a handful of highly localized ground measurements scattered across, mostly, the northern hemisphere land surface; ...":

Strictly speaking this is not true. The first space-based instrument measuring XCO2 was SCIAMACHY on ENVISAT (Bovensmann et al., 1999), launched already in 2002, and SCIAMACHY XCO2 retrievals have been used to study the carbon cycle already before (e.g., Buchwitz et al., 2007; Schneising et al., 2008) but also after (e.g., Reuter et al., 2014; Schneising et al., 2014) the launch of GOSAT. This information needs to be added.

**A1:** Not sure why SCIAMACHY was not referenced in this discussion. The text has been revised, see lines 25-34:

"Since the launch of the SCanning Imaging Absorption spectroMeter for Atmospheric CartographY (SCIAMACHY; Bovensmann et al., 1999) in 2002 aboard the European Space Agency's Envisat, space-based instruments have been addressing the particular challenge of scale. In decades prior, the global carbon cycle was studied using a handful of highly localized ground measurements scattered across, mostly, the northern hemisphere land surface; SCIAMACHY and its successors have changed this limitation profoundly. The Greenhouse gases Observing Satellite (GOSAT; 30 Kuze et al., 2009; Yokota et al., 2009) launched in 2009, as well as the Orbiting Carbon Observatory missions (OCO-2 and OCO-3), both launched in the 2010s, have improved upon SCIAMACHY's ability to measure CO2 over a large fraction of Earth's surface, with more continuous spatial coverage than ground-based networks can provide. With their increasingly fine spatial resolution, precision, and accuracy, space-based observations from the OCO missions can now resolve carbon sources around the globe on scales as fine as those of individual power plant plumes (Nassar et al., 2017)."

**Q2:** Please add more information and if possible also references on the "challenge of scale" aspect. What exactly is the challenge? Where has it been addressed?

**A2:** The phrase "challenge of scale" refers to the proceeding sentence – simply the spatial limitations of ground-based networks.

**Q3:** Page 2, line 50: Sentence "… producing a data-dense, spatially coherent map of XCO2.":

As the maps shown in the paper indicate that the OCO-3 SAM XCO2 product suffers from significant biases I would conclude that the goal of generating "spatially coherent map of XCO2" has not yet been achieved. I recommend to write "aims to produce" (or equivalent) instead of "producing".

**A3:** It is valid to suggest that a producing a "spatially coherent map of XCO2" is still a work in progress, especially in light of the significant geolocation errors in OCO-3 vEarly. On lines 51-53, the text has been changed to read:

"In Target and SAM modes, the instrument points at a specific off-nadir location and scans multiple times during an overpass, in an effort to produce a data-dense, spatially coherent map of $XCO_2$ . "

**Q4:** Page 2, lines 56-57, sentence "Point source signals are difficult to quantify because the XCO2 enhancement is often two orders of magnitude smaller than the background concentration":

The difficulty does not arise from the fact that the enhancement is two orders of magnitude smaller than the background concentration, but from the fact that the instrument noise is about the same order of magnitude as the enhancement.

**A4:** Regarding instrument noise being the same order of magnitude as a typical point source enhancement, on lines 58-60 the text has been changed to read:

"Point source signals are difficult to quantify because the instrument noise is a similar order of magnitude to the $XCO_2$ enhancement; the $XCO_2$ enhancement is also often two orders of magnitude smaller 60 than the background concentration. "

**Q5:** Figure 1: According to the figure caption the left figure shows "a power plant plume". Visible in wind direction are two areas of elevated XCO2 (instead of a single plume area). Is it possible to comment on this? Is this supposed to be a real feature or a bias related artefact? Interestingly the TROPOMI NO2 figure on the right shows something similar (despite the time difference).

**A5:** In Figure 1 we believe the XCO2 enhancement(s) to be a real feature, based on how well it compares to the TROPOMI NO2 data. Further investigation as to the reason for the two separate areas of enhancement is not relevant to the purposes of the figure within the discussion, but we have clarified the pattern in our phrasing. Lines 55-65 now read:
"Figure 1 provides an example of a visible $X_{CO_2}$ enhancement (or, rather, two areas of enhanced $X_{CO_2}$ ) over a power plant site, as seen by OCO-3. In this case, the enhancement extends to the northeast across four OCO-3 swaths, which we define as individual along-track scans. Each swath is eight footprints wide - these are visualized in the right-hand panel as white rectangles. Point source signals are difficult to quantify because the instrument noise is a similar order of magnitude to the $X_{CO_2}$ enhancement; the $X_{CO_2}$ enhancement is also often two orders of magnitude smaller than the background concentration. $NO_2$ concentrations are a helpful validation source for fossil signals due to $NO_2$'s short lifetime and high concentration relative to background values. We show observations from the Tropospheric Monitoring Instrument (TROPOMI) $NO_2$ product (Veefkind et al., 2012; Van Geffen et al., 2019) in the right-hand panel of Figure 1. Because the OCO-3 $X_{CO_2}$ and TROPOMI $NO_2$ observations compare so well, we believe the $X_{CO_2}$ enhancement to be a real feature of the atmospheric state in this scene. "

**Q6:** Page 7, equation (1):

The interpretation of sb_ratio as "swath bias" assumes that the (real) CO2 plume is negligible in terms of amplitude and/or area coverage, or? If yes, then I recommend to add this information.

**A6:** It is true that our definition of sb_ratio assumes that the real CO2 plume is negligible. We explore the effects of this assumption via our subjective analysis of v10 SB cases in Section 7, where we find that this interpretation of sb_ratio does leave room for error, i.e., flagging real CO2 plumes or other across-swath biases as "swath bias." The text has been revised to reflect this.

See lines 188-191 in Section 3:
"We acknowledge that this interpretation of sb_ratio assumes that any sufficient across-swath $X_{CO2}$ variability is due specifically to SB, meaning that real $X_{CO2}$ signals or other biases are negligible; this leaves room for error in our interpretation of the sb_flag, which we explore later in the Version 10 data product in Sect 7. "

See also lines 580-582 in Section 7:
"We acknowledge that vEarly suffers the same limitations as v10 in terms of this interpretation of the sb_flag, indicating that vEarly likely also suffers a far smaller number of "true" SB cases than our flag suggests. Despite this, we still consider v10 an improvement based on the across-swath and within-swath improvements to the $X_{CO2}$ field shown in Figure 16. "

**Q7:** Page 8, line 186 and following: The dP_abp filter seems quite relaxed as a 16 hPa surface pressure error corresponds roughly to 1.6% or 6 ppm for XCO2. It is written that this is part of "a simple post-processing quality filter". In the previous paragraph it is written that ABP is part of the pre-processing. I find this confusing. Is the filter used for post-processing but computed already during A-band pre-processing? Is this a difference between the operational retrieval algorithm and the one used here? Please clarify. Is the pressure difference dP also computed using the retrieved state vector elements (if surface pressure is a state vector element) originating from the main (3-band L2FP algorithm) retrieval and if yes is this (L2FP) dP also used for quality filtering and bias correction? And if not, why not?

**A7:** See lines 193-194:
"The filtering process for the OCO missions in general involves two stages: pre-processing, which eliminates soundings prior to the L2FP retrieval (also called "sounding selection"), and post-processing,"

and lines 205-210 for clarification:
"In the sounding selection process, a |dP_abp| < 30 filter is used to eliminate cloudy soundings before pushing clear" soundings through to L2FP (Taylor et al., 206). We choose in this study to use an additional post-processing simple filter of |dP_abp| < 16 hPa to define "good" quality soundings. A similar range of dP_abp is typically used in the development of the operational quality flags (Taylor et al., 2020) This alone acts as a fairly relaxed filter, but it is critical for our investigation to retain enough soundings to see the swath bias. For the rest of this study, we will refer to this as the dP_abp quality filter."

**Q8:** Page 8, line 205. Figure 5c is mentioned, but there is no figure 5c, because figures have no (a), (b) ,..., labels. The "f" in "figure" should be capitalized.

**A8:** "Figure" has been capitalized on line 225, and Figure 5 panels have been labeled with (a), (b), etc.

**Q9:** Page 11, section 5:

I understand that the retrieval algorithm/code as applied to the simulated $XCO_2$ data is exactly identical with the algorithm/code used to analyse the real OCO-3 data apart from different spectroscopic input data. Or are there any significant (other) differences (including pre- and post-processing)?

**A9:** No, apart from our use of dP_abp as a filter rather than the operationally included xco2_quality_flag

**Q10:** Figure 6 and related discussion (including Figures 10 and 15):

The simulated $XCO_2$ as shown in the top right panel shows a large discontinuity – a $XCO_2$ "jump" of several ppm between the "4 bottom left swaths" and the "4 top right swaths". It is concluded that: "We find that simulated spectra derived from simple aerosol scenes are successfully able to generate SB patterns similar - though not identical - to those in the operational vEarly data". Yes, but why? This is not clear for me and I find this very surprising. I would have expected a more smoothly varying bias assuming that neither the surface properties nor the aerosols show a corresponding jump. Is this assumption true for the simulations (I assume that maps of the relevant input parameters have been generated and investigated)? Which input parameter as used for the simulations shows a similar jump and can therefore explain the $XCO_2$ jump? If all input parameters vary smoothly than the result indicates that the retrieval algorithm seems very sensitive to small changes of certain input parameters as used for the radiative transfer simulations to generate the simulated spectra. In this case, for some reason, the retrieval responds with a jump from one state to another, which is a

bit unexpected. I recommend to generate and inspect maps of relevant input parameters (in particular also viewing angles) which may explain the jump including parameters such as the relative azimuth angle between line-of-sight and sun direction which may also jump / change sign. In this context: I assume that the swaths are not parallel to the flight direction and the "4 bottom left swaths" are not on one side of the sub-satellite track and the "4 top right swaths" are not on the other side, or? In any case please add information on how the scans are performed in terms of timing (I assume that there is only a small time difference between the different swaths and that one swath after the other (from left to right or the other way around) is measured).

The results shown in the bottom right panel of Figure 6 are even more surprising as the between swath jumps are even less systematic also suggesting the issues may be related to certain angles (assuming that none of the other input parameters is spatially correlated with the swaths).

Figure 10 shows that the retrieved XCO2 (from the simulations) significantly "jumps" depending on the assumed aerosol type with more or less large XCO2 jumps within the scene. Again, this is surprising if surface and albedo properties are not spatially correlated with the XCO2 bias pattern.

**A10:** Section 6 has been altered to better explain the nature of the swath bias. We have added four figures, which replace the previous Figure 6, which show via a series of maps that the primary driver of large changes on a swath-to-swath basis is the viewing geometry. While some other retrieved parameters may vary between swaths, it is difficult to see purely geometry-driven effects in the operational data due to variability from other aspects of the scene, such as real aerosol heterogeneity, topography, or surface albedo heterogeneity. We show that when those additional sources of variability are held constant in our simulations, the geometry-driven effects become apparent in the other retrieved parameters, not just XCO2.

We have also added brief additional discussion to Sections 6.2 and 6.6 to enforce this point. See lines 397-401:
- "However, in terms of SB orientation, the May 8 case illustrates particularly well the fact that different aerosols produce different geometry-dependent responses: the location of the highest and lowest XCO2 values occurs in different swaths depending on the aerosol type. This makes sense given the unique optical properties of each aerosol type, but would require further study to predict with quantitative skill. We conclude that the physics of SB, in terms of both magnitude and direction, are highly dependent on the aerosol type, and are complex enough to warrant further study."

- Lines 462-464: "In our simulation work, we show that the large changes in XCO2

between swaths are primarily correlated to the changing viewing geometry, and that by eliminating other sources of variability within a SAM, we can simulate the purely geometry-driven response within the retrieval."

- Lines 479-481:"Biases or real variations in XCO2 derived from additional complexity in real scenes can also enhance this signal - varying surface albedo or topography, for instance."

Finally, we have added a few statements to reflect this in Section 8 (Conclusions):
- Lines 607-621: "By studying real OCO-3 SAMs, we show that SB in XCO2 is primarily correlated to the viewing geometry, though the geometry signal in other retrieved parameters appears easily obscured by signals from other sources, such as heterogeneous aerosol fields or surface topography. To remove additional in-scene variability, we build custom aerosol and surface scenes to generate simulated radiance spectra, and use those spectra to retrieve XCO2 with the ACOS L2FP algorithm. Our first tests show that we can successfully reproduce geometry-driven SB effects via this method, and while not identical to the observed vEarly SB, the similarities are enough to instill confidence in the utility of the procedure. We select a set of three SAMs over an Australian desert site, which represent a range of solar zenith angles and viewing geometries, as well as a range of SB effects. For these three SAMs, we perform a series of controlled tests, changing individual inputs to the simulated scene. [... ] Scenes with no aerosol show little to no SB, but each SAM's response to aerosol height and optical depth is consistent - the higher the aerosol within the column, or the higher the AOD, the stronger the SB. We test one coarse and one fine mode aerosol for each SAM, and each produces a different SB, but which is stronger depends on the conditions of the SAM. The precise combination of geometry and aerosol optical properties required for a strong SB appears to be complex, although it is apparent that the unique optical properties of each aerosol type interact with the geometry to produce the different XCO2 patterns. The exact nature of these interactions would require further study to describe in a quantitative manner."

**Q11:** Figure 15 shows that dP also "jumps", i.e., shows a spatial pattern correlated with the XCO2 bias. Is the dP shown in the bottom left panel computed with the operational ACOS ABP algorithm (using only the O2-A-band) or is it computed using output from the 3-band L2FP retrieval? It seems that the XCO2 jumps are strongly related to dP jumps (and therefore using dP for bias correction helps to reduce biases). As dP originates (entirely or mainly) from the O2-A-band then the question is if the origin of the XCO2 biases is related to the use of the O2 A-band (as part of the 3-band L2FP

retrieval)? Can it be excluded that the use of the O2-A-band causes the presented XCO2 biases (in particular the XCO2 jumps)?

**A11:** The dP map in Figure 15, as well as the new Figures 6, 7, 8, and 9, is output from the the L2FP retrieval. Our response to reviewer 2's question about the swath bias being driven by dP (similarly, by the O2 A-band) addresses this question: the summary is, if dP were driving the swath bias, we would expect v10's updated bias correction to have a large effect on the number of swath bias SAMs. In fact, we find that the quality filtering has a more substantive effect, so we believe the swath bias to be driven more by aerosols generally not being characterized properly in the retrieval than by our surface pressure retrieval. See the response to Reviewer 2, **Q1/A1** for the more detailed explanation.

**Q12:** Page 19, line 405-406, sentences: "we surmise that the trend toward SB at higher albedos is not because SB is more likely to occur over bright scenes. In fact, stronger SB tends to occur at lower albedos" but "The SB is highest at lower albedos" (lines 398-399):

Which trend toward SB at higher albedos is this referring to?

**A12:** Revised lines 455-456 now read:
"From this exercise, we surmise that the trend toward SB at higher albedos observed in Sect. 3, Figure 5 is not because SB is more likely to occur over bright scenes."

**Q13:** Page 22, lines 468-469: "we first apply the more restrictive v10 sounding selection criteria to vEarly":
Do these selection criteria refer to the quality filter? Because right after that sentence: "We apply quality filtering and bias correction, narrow down to only SAMs with at least 500 soundings (N>500), and calculate our SB parameters from Equation 1. vEarly quality filtering is our custom | dP_abp | < 16 hPa filter, and v10 is filtered using the operational v10 quality flags. Figure 16 details the comparison". Which filter is used for vEarly? According to results apparently the custom | dP_abp | < 16 hPa filter.

**A13:** "Sounding selection" is the application of pre-processing quality filters and other basic checks on the L1b (observed radiance spectra) results. See response to earlier question regarding the dP_abp filter.

**Q14:** Figure 16, caption: the histograms seem to be normalized. I would add this information. The abbreviation NSAM (in the figure) is not explained.

**A14:** Lines 525-527 revised:
"We apply quality filtering and bias correction, narrow down to only SAMs with at least 500 soundings ("N>500" representing the number of soundings, "NSAM" representing the number of N>500 SAMs), and calculate our SB parameters from Equation 1."

Figure 19 caption revised:
"Normalized histograms of swath bias parameters for vEarly and v10 SAMs, using v10 sounding selection for both datasets. All SAMs have at least 500 soundings (N>500). The number of SAMs in each dataset is given by NSAM. vEarly is in blue, v10 is in orange, and the SB threshold of sb_ratio > 0.75 is indicated by the gray line. The number of swath bias cases is listed as NSB in the right-hand panel."

**Q15:** Page 23, line 481: "Bias correction alone reduces the frequency of v10 SB cases from 11.9 to 10.4%": These numbers do not match the NSB/NSAM shown in Fig. 16:

(vEarly) 378/2685 = 0.14

(v10) 202/1749 = 0.12

Do the percentages refer to something else? The 11.9% matches the numbers mentioned in page 8, lines 191-192. If this is the case, the total set of SAMs for the comparison is not the same.

**A15:** These numbers refer to analysis not shown in Figure 16, in which we apply v10 bias correction and quality filters each individually before applying them together. This particular statistic has been removed, to avoid confusion. Lines 538-540 now read: "Quality filtering and bias correction each have their own effect on the swath bias, and improvements due specifically to the v10 bias correction could be derived either from the improved dP correction or the  [...]"

**Q16:** Page 25, line 512: I recommend to add NO2 after TROPOMI: TROPOMI NO2 indicates ...

**A16:** Text has been updated in Line 573:
"TROPOMI NO2 indicates a real X$CO_2$ plume, [...]"

**Q17:** Page 26, caption Figures 19 and 20: Please add info on which product is shows in which panel. Is the product shown in the middle the "Lite" product?

**A17:** The title of each plot provides the details of the data: the left-hand panel is the raw XCO2 data from the lite file (no bias correction and no quality flags), the middle panel is the XCO2 with bias correction and quality flags apsplied.

**Q18:** Page 26, line 530, sentence "… we observe a new geometry-related bias …": This sounds that it can be excluded that OCO-2 retrievals also suffer from this bias. As the OCO-3 data are similar as the OCO-2 data and also the retrieval algorithm is essentially the same I am not sure that this is really a new bias in the sense that only OCO-3 data suffer from it. Have similar issues (especially XCO2 jumps) also been observed for OCO-2 (e.g., target mode observations)?

**A18:** OCO-2's pointing mechanism is entirely different from OCO-3's when it comes to Targets versus SAMs. OCO-2 adjusts the geometry of the satellite, whereas OCO-3 relies on changes made to the pointing mirror assembly. OCO-2 Targets also cover a much smaller area; the variation in geometry is generally much smaller than in SAMs. We hypothesize, based on these two factors, that the swath-dependent bias is OCO-3 specific. We would also assume that bias correction and quality filtering help remove those effects in OCO-2, as they do in OCO-3 v10. However, any bias-inducing interaction between aerosols and surface properties/viewing geometry within the retrieval will be present in OCO-2 data, though likely smaller in magnitude.

**Q19:** Page 26, line 532 following, sentence "… by calculating the ratio of swath-to-swath noise in the XCO2 field to the …": This quantity is referred to as swath bias in the paper as it is a systematic error and not a random error, i.e., not noise. I recommend to replace "noise" by "scatter" or "standard deviation of the medians computed for each swath" or equivalent.

**A19:** Lines 598-600 revised to use the word "scatter":
"by calculating the ratio of swath-to-swath scatter in the $X_{CO_2}$ field to the scatter over the full scene and triggering a swath bias flag (sb_flag) over a threshold value of 0.75 (Equation 1). "

**Q20:** Page 27, line 535 following: Why are so many fossil targets suffering from swath bias? Can this be an artefact of the analysis as the computation of the indicator (see Eq. 1) assumes negligible plumes?

**A20:** We believe this to be a combination of the higher AODs typically observed over fossil sites *and* the assumption of negligible plumes.

**Q21:** Page 27, line 561, sentence "Finally, we replicate our vEarly analysis using the updated version 10 dataset, and see vastly improved statistics. We find that improved quality filtering is the primary driver of this development, …": I guess that "vastly improved statistics" primarily refers to relative (percentage) performance (as filtering removes data) but not to absolute performance in terms of also more good data. Please extend this statement so that it is clear if also the absolute number of "good" retrievals is enhanced or not.

**A21:** We believe the proceeding statement accurately describes the improvements to the data quality: "Better sounding selection and significant geolocation improvements, combined with a better dP bias correction, also make v10 more effective at mitigating SB effects, and generally improve the quality of the final data product."

**Typos etc.:**

Page 8, line 205: "of important" -> "of importance" (or equivalent)
Revised line 225, reads "of import"

Page 12, line 295: representative
Fixed, on revised line 317: "a single representative target location"

Page 19, line 407: "aerosols and dark surfaces" → "aerosols over dark surfaces"
Fixed, see revised line 457: "differentiating aerosols over dark surfaces, whereas it has trouble identifying […]"

- https://doi.org/10.5194/amt-2022-241-RC2

REVIEWER 2

**Q1:** The ACOS retrieval uses the retrieved surface pressure to calculate XCO2. I am wondering how much trouble actually comes from the surface pressure retrieval (mostly informed by the O2A band) and how much from the actual CO2 retrieval. The large effect of the change in "dP correction" on the SB (section 7) fuels my concerns. Also, the fact that coarse aerosols (with a smooth spectral variation of optical properties between O2A and strong CO2 band) seem less problematic than fine aerosols (with a substantial spectral variation) could hint at particular difficulties in getting a "consistent scattering picture" from the O2A and CO2 bands. Would it be possible to separate the surface pressure related portion of the SB? How large is it – if it is large, why not use a priori surface pressure?

**A1:** We have explored using the a priori surface pressure, in experiments leading to up v10. Various surface pressure constraints were tested and the results evaluated after bias correction. The results of using the prior surface pressure, among other tests, were consistently not as good as retrieving the surface pressure and then bias correcting it out after the fact – the reasons for this are unknown.

Regarding dP as a driving force of the swath bias, the best evidence we have addressing this question thus far is our evaluation of the v10 dP bias correction. If dP were a driving force of the swath bias, we believe that applying the v10 bias correction would decrease the number of v10 swath bias cases significantly. We have added text to lines 548-552 addressing this:
"We do note, however, that in tests applying the bias correction and quality filtering separately, the quality filtering had the more substantial effect on the SB: bias correction alone reduced the number of v10 SB SAMs from 325 (in raw data) to 310, and quality filtering reduced it from 325 to 225. While dP had the largest impact within the bias correction, the quality filtering had an even larger impact, indicating that the swath bias is not driven specifically by dP, but rather by extreme aerosol effects generally being characterized poorly within the retrieval."

**Q2:** While I like the approach to concentrate on understanding individual scenes, I find the focus on one particular scene in Australia quite narrow. The scene is bright and surface reflectivity is probably spectrally smooth throughout the spectral range covered. This implies that the dominating scattering effect in all bands (somewhat depending on geometry) is light path enhancement due to (multiple) reflections between ground and aerosol layer. While the authors touch on the effect of surface albedo (Fig. 13), I would recommend examining in depth another, darker scene with substantial spectral variation in surface albedo (e.g. vegetation). For darker scenes, light path shortening due to direct backscattering from the aerosol layer would be more important i.e. discussion of such a scene would cover an entirely different radiative transfer regime and thus, it could contribute mechanistic understanding.

**A2:** We have changed the way we introduce our experimental setup in an attempt to address the concern regarding the limitation of the study to a single site. See lines 307-310:
"We then focus on three SAMs over a single representative target location, and use their geometry as templates to test the SB in a more complete scene state space: [..]"
and lines 332-335:
"By examining three SAMs from the same site, we are able to investigate the differences in atmospheric state and/or observation geometries that drive the operational SB, in addition to using their different geometries as a template for a broader array of synthetic scenes, as mentioned above."

While we choose a single scene over Australia as the basis for our simulations, rather than focus solely on the real scene from each of the three dates chosen, we use the geometry of each date to test a broader state space - including albedos ranging 0.1 to 0.6. We also feel that focusing on scenes over bright surfaces is warranted because our vEarly analysis (see Figure5d-f) indicates that swath bias is a more acute problem in SAMs with high surface albedo.

**Q3:** L165f: I got quite a bit confused with the directions „across swath" and "along swath". I understand that a simple "across/along track" does not work because OCO-3 has a dedicated pointing system such that the scanning is not aligned with forward direction of the space station. Maybe the authors could consider to make a small sketch defining their notations or include the notation in one of the early figures.

**A3:** Visualization of this language is now included in Figure 1: we identify individual swaths, and indicate the along- and across-swath directions.

**Other minor changes:**
(1) "We apply our single profile, along with its associated surface elevation and surface reflectivity, to every sounding in the SAM." Added to Section 4.1. The constant elevation/reflectivity was not specified previously.
 - similarly, in Section 6:

   "[…] we manipulate each SAM to include various aerosol types, heights, and optical depths with *realistic surface*"

  changed to

   "[…] we manipulate each SAM to include various aerosol types, heights, and optical depths *with a constant surface elevation and reflectivity*" to better reflect the simulation setup.